# A Review of Advances in Cold Spray Additive Manufacturing

Rodolpho Fernando Vaz * , Andrea Garfias, Vicente Albaladejo , Javier Sanchez and Irene Garcia Cano

Thermal Spray Centre CPT, Universitat de Barcelona, 08028 Barcelona, Spain
* Correspondence: rvaz@cptub.eu

**Abstract:** Cold Spray Additive Manufacturing (CSAM) produces freeform parts by accelerating powder particles at supersonic speed which, impacting against a substrate material, trigger a process to consolidate the CSAM part by bonding mechanisms. The literature has presented scholars' efforts to improve CSAM materials' quality, properties, and possibilities of use. This work is a review of the CSAM advances in the last decade, considering new materials, process parameters optimization, post-treatments, and hybrid processing. The literature considered includes articles, books, standards, and patents, which were selected by their relevance to the CSAM theme. In addition, this work contributes to compiling important information from the literature and presents how CSAM has advanced quickly in diverse sectors and applications. Another approach presented is the academic contributions by a bibliometric review, showing the most relevant contributors, authors, institutions, and countries during the last decade for CSAM research. Finally, this work presents a trend for the future of CSAM, its challenges, and barriers to be overcome.

**Keywords:** cold spray; additive manufacturing; 3D-printing; geometries; properties; innovation

## 1. Introduction

Additive Manufacturing (AM) has been an industrial revolution in recent decades, starting with producing polymeric parts and advancing to metallic components. Many alloys and methods have been studied, some more industrially mature and others in a developing stage. The definition of Additive Manufacturing (AM) given by ISO/ASTM 52,900:2015 standard [1] is the "process of joining materials to make parts from 3D model data, usually layer upon layer, as opposed to subtractive manufacturing and formative manufacturing methodologies". Other nomenclatures have been used worldwide as synonyms for AM, such as 3D printing, additive fabrication, rapid prototyping, and others. AM has been used to build prototypes, manufacture the final products, or even repair damaged components, innovating the global manufacturing industry [2–6]. Many companies have invested in developing new AM techniques and materials, optimizing the process parameters, reducing costs, and making the AM a competitive piece of technology [7,8]. Different sectors have benefited from using AM [9,10], such as medical [11–16], aerospace [17–20], automotive [21–23], supply chain [6,24–26], and others. Compared to the traditional subtractive manufacturing techniques, AM is characterized by being less wasteful, enhancing resource efficiencies, and changes in the design and production phases. Kozoir [27] presents the effectiveness of optimizing AM processing parameters to reduce the mass of models, keeping the desired mechanical properties. AM also extends the product life cycle by repairing high-cost parts, and reconfigures the value chains to be shorter, collaborative, and offer remarkable sustainability benefits [6,28]. In this way, AM offers clear benefits from the viewpoint of sustainability [29–31].

The commercial use of AM emerged for polymers in the 1980s, introducing Stereolithography (SL), which involves curing a photosensitive liquid polymer by a laser beam [32,33]. An evolution in equipment changed the raw material to the powder form, using Selective Laser Sintering (SLS) to fuse this powder [34]. Other classes of AM for polymers are Material Jetting (MJ) [35,36], Binder Jetting (BJ) [37,38], Material Extrusion

(ME) [39,40], and Sheet Lamination or Laminated Object Manufacturing (LOM) [41]. The techniques consolidated for polymers have been successfully applied for other materials also, such as BJ for ceramics and metals [37,42,43], LOM for metals [44,45], and ME for composites [40,46]. Various processes are available for metal AM processing for the most different alloys and applications. The selection or choice of the adequate process depends on the part's geometry, complexity, mechanical properties, and other factors [47,48].

The metal AM processes differ from the heat source and metal feeding method or type. Some options are the laser process, Selective Laser Melting (SLM) or Sintering (SLS), Direct Metal Laser Melting (DMLM) or Sintering (DMLS), or Laser Metal Fusion (LMF), besides the Electron Beam Melting (EBM) process [49–51]. These are methods which are applied to the parts that need low or no machining post-processing or are used directly as end-use products. Other processes are presented in the literature but are not capable of producing complex geometries, such as Gas Tungsten Arc Welding (GTAW) [52–54], Gas Metal Arc Welding (GMAW) or Wire Arc Additive Manufacturing (WAAM) [55–58], Plasma Arc Welding (PAW) [57,59–61], Friction Stir Energy Manufacturing (FSAM) [62,63], and Ultrasonic Additive Manufacturing (UAM) [64,65]. Examples of AM by welding processes that demand post-machining are repairing long fatigue cracks in hydro powerplant runners [66] or repairing eroded gas turbine blades [67].

Cold Spray (CS) is a thermal spray process designed for coatings that has extended its use to produce freeform parts [28,68–70]. CS produces harder microstructures than other AM processes, as studied by Gamon et al. [71], who present CSAM-ed Inconel 625 with 600 HV. On the other hand, WAAM, SLM, EBM, DMLM, and BJ resulted in less than 300 HV. Figure 1 presents the AM technology maturity, evidencing the actual industrial use of the laser processes, SLM and DMLM, as WAAM. The prediction is to use CSAM industrially in a short time, less than two years, but a long development journey for FSAM and UAM [72]. This work aims to present the trodden path by CS as an AM technique and the foreseen way to consolidate and diffuse CSAM in the industry. Figure 2 shows examples of AM-made products employing different strategies.

This paper presents and discusses the evolution and advances of CSAM critically, following the scheme shown in Figure 3.

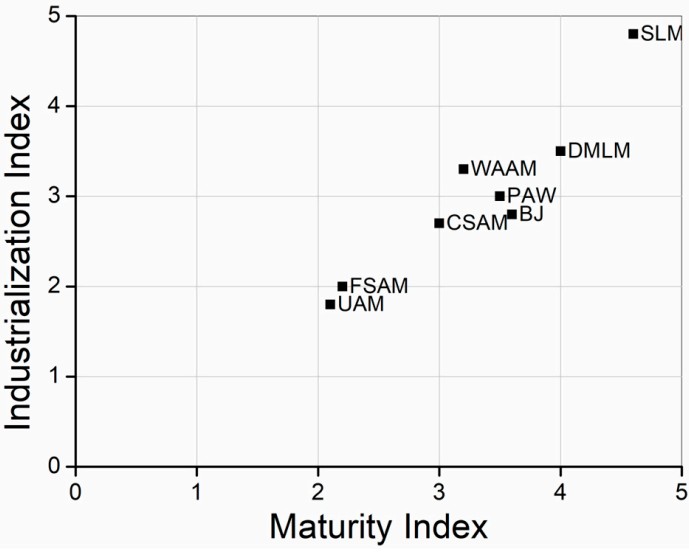

**Figure 1.** AM maturity index for producing metallic parts.

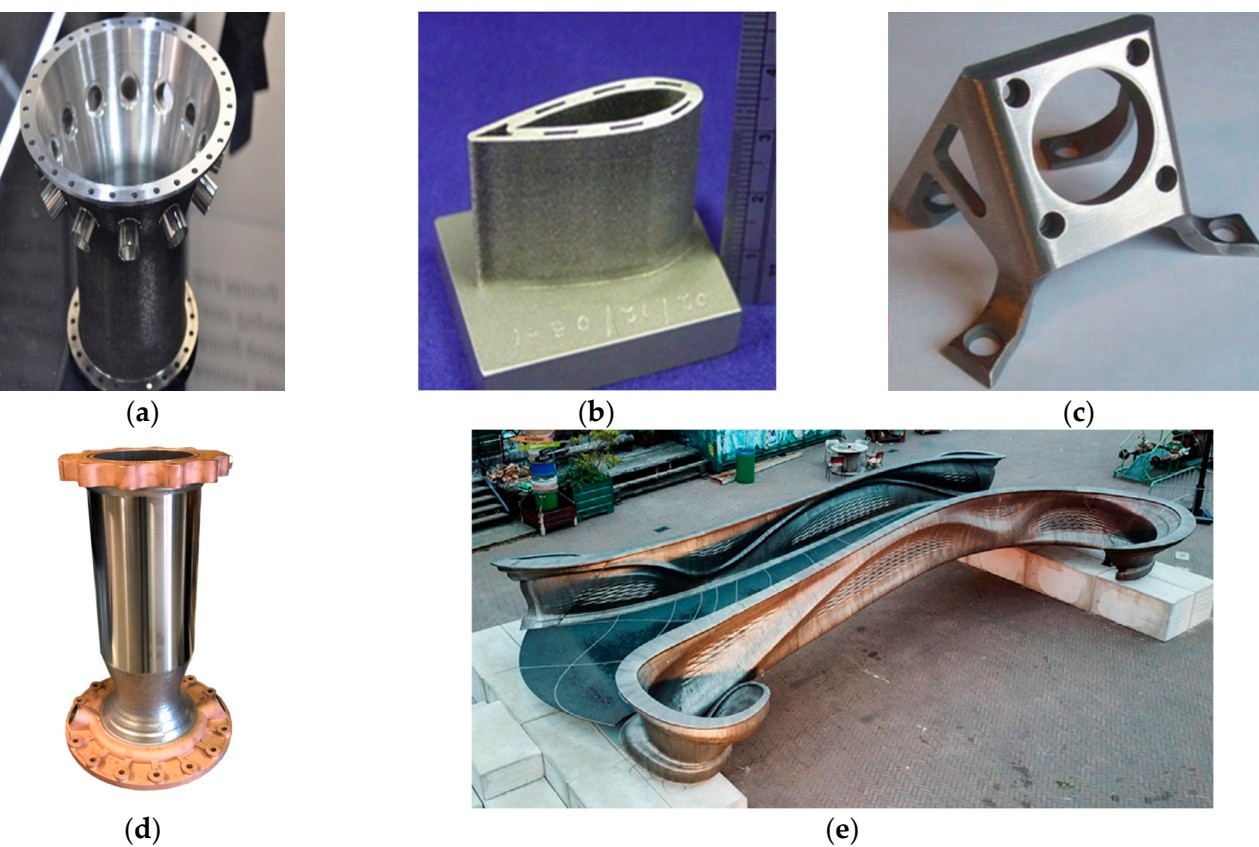

**Figure 2.** Metal AM parts. (**a**) DMLM-ed stainless steel gas turbine housing. Reprinted with permission from Ref. [18], Elsevier, 2017. (**b**) DMLM-ed Ti6Al4V airfoil [18], (**c**) CSAM-ed Ti bracket. Reprinted with permission from Ref. [2], Elsevier, 2018. (**d**) CSAM-ed and DMLM-ed bimetallic thrust chamber. Reprinted with permission from Ref. [73], NASA, 2021. (**e**) WAAM-ed stainless steel bridge in Amsterdam. Reprinted with permission from Ref. [74], Elsevier, 2019. Unit: mm.

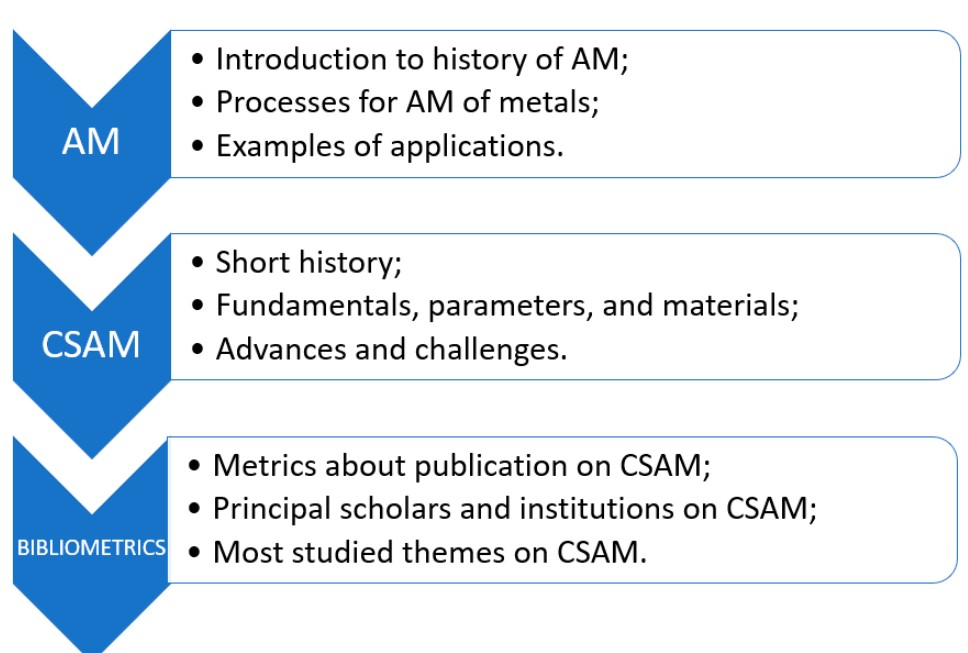

**Figure 3.** Flowchart of the topics presented in the work.

## 2. Cold Spray Process

This section describes the CS process, its fundamentals, principles, parameters, and their selection, which is essential to understanding and placing CS as an AM process. CS is a thermal spray process investigated and presented by many authors as an alternative to producing AM freeform parts. It has severe differences from the laser, welding, and other thermal spray processes since CS does not change the properties of the feedstock powder by heating or melting during the AM part fabrication because the powder is kept below its recrystallization temperature during the spraying time [75–77]. However, CSAM produces parts with a very high density, >99%, due to the very high velocity imposed on the particles, reaching supersonic velocity values [78–80]. Therefore, the correct selection of feedstock powder, deposition parameters, and strategy are fundamental for achieving high Deposition Efficiency (DE) and good CSAM performance [81–84]. CS also prevents materials oxidizing during the deposition due to the relatively low temperature that the material absorbs at the spraying time [85]. In addition, CS avoids other harmful effects seen in other AM or thermal spray processes, such as evaporation, melting, recrystallization, tensile residual stresses, debonding, and gas releasing, besides the ability to deposit high-reflective metals such as Cu and Al [28,83,86,87]. A great CSAM advantage is the possibility of the deposition of dissimilar materials, e.g., a sandwich-like structure of Cu and Al [88], which is not feasible by welding.

Historically, CS has been presented in the literature by different names: "kinetic energy metallization", "kinetic spraying", "solid-state deposition", or "high-velocity powder deposition", and others [78,89]. Its principle and physics of operation were studied during the XX century, with the operational evolution starting in the 1980s. Still, its commercial development started just in the early 2000s [86,89,90], increasing its expansion from the R & D sector to the industry since then, and with a prediction of widespread industrial use in less than two years [91]. CS is the thermal spraying process to which a large number of studies and publications have been devoted over recent years, presenting its principles and physics, but, nowadays, emphasizing industrial or real applications of the technique and mainly its use in the AM field [2,28,81,92–97]. The monetary benefits are imperative to select CSAM as an industrial production technology. A comparison among the metal AM technologies was presented by Munsh et al. [91], and CSAM was highlighted as the lowest cost per volume fabricated and the highest deposition rate, reaching $kg \cdot h^{-1}$ [79,81]. Besides the component at hand, the advantages of AM over traditional or subtractive fabrication processes include the redesign potential of the whole system, which is not easily measurable [91].

CSAM produces a coating or bulk component generated by a solid-state cohesion during the powders' impact on a substrate. The working gas is previously heated in a chamber, reaching high pressure, flowing through a de Laval or similar convergent–divergent nozzle, accelerating it to supersonic velocities, and dragging the feedstock powders. [68,78,98]. The working gas pressure classifies CS, as presented schematically in Figure 4. Low-Pressure Cold Spray (LPCS) operates under 1 MPa, and High-Pressure Cold Spray (HPCS) uses higher pressure levels. A Medium-Pressure Cold Spray (MPCS) is a commercially available system, Titomic D623. LPCS is limited to a few materials and can be portable or manually operated, accrediting it for in-field operation and repair services. At the same time, HPCS is the CSAM used for many materials, but has heavier and more equipment than LPCS, employing a bigger gun, heat exchanger, energy source, robot, and acoustic enclosure (soundproof booth) for the operation, because the noise usually exceeds 100 dB [2,10,99,100]. This change in gas pressure and equipment configuration influences the sprayed particle velocity since the high velocity of particles is a consequence of high gas pressure and the nozzle design [101–105]. Another difference between the LPCS and HPCS is the powder feeding; for the first one, the particles are dragged by the working gas in the nozzle directly, using a downstream mode. On the other hand, HPCS uses an upstream injection mode, and the powder feeder is connected to a feeding gas line,

which improves the powder flowability, and increases the range of powders which are CS sprayable [90,106].

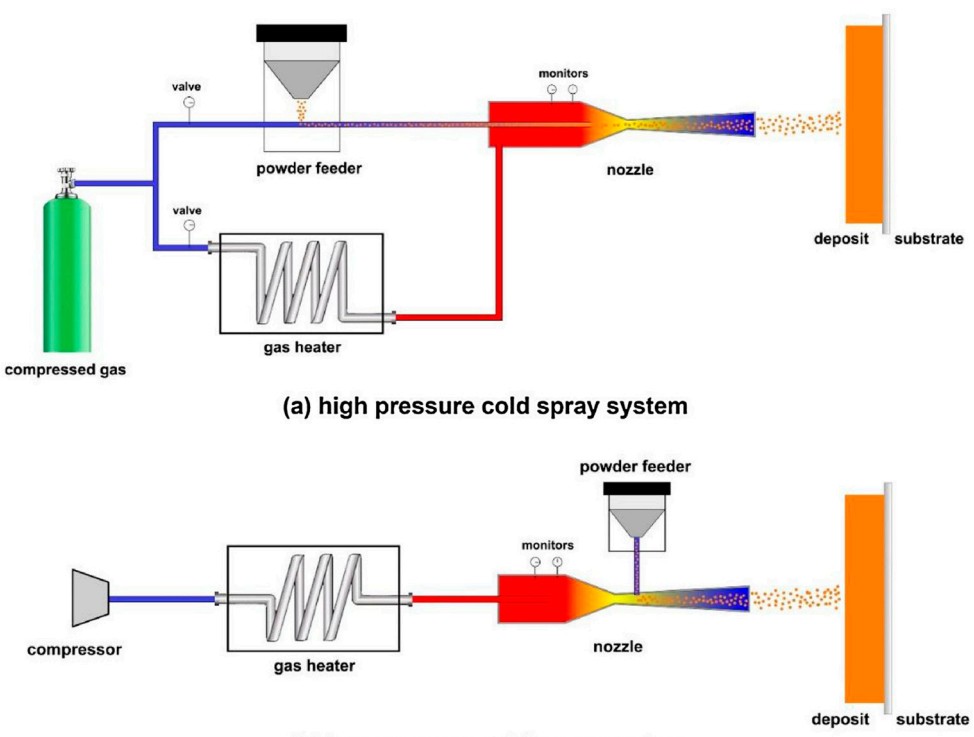

**Figure 4.** LPCS and HPCS schemes. Reprinted with permission from Ref. [2], Elsevier, 2018.

The bonding mechanism of the solid-state particles to a substrate still has to be understood entirely. However, it is believed that their high energy at the impact disrupts the oxide films on the particle and substrate surfaces, pressing their atomic structures into intimate contact with each other under short high interfacial pressures and temperatures. This mechanism is called Adiabatic Shear Instability (ASI) [107,108]. It supports the success in coating ductile materials, such as Cu and Al, and flops in spraying brittle materials, such as ceramics or carbides [78]. At the impact, most of the kinetic energy from the in-flight particles is converted into heat or the plastic deformation of the substrate and the particle, which can produce strain, ultimately shear instability, and jetting. With an increase in local temperature, thermal softening alters the capacity of the material to transmit shear forces, and eventually, the softening process dominates over strain hardening [109–111]. Hassani-Gangaraj et al. [112] show the jetting happening with or without the material having a thermal softening capacity, proposing that CS jetting is formed as a result of strong pressure waves in the particles, expanding the particle edges. This mechanism is related to hydrodynamic processes that promote jetting, such as liquid droplet impacting, shaped-charge jetting, and explosion welding. The critical velocity ($V_{cr}$) of particle for the bonding was mathematically related to the bulk speed of sound, which was minutely commented by Assadi et al. [113], who refuted those conclusions and sustained the ASI as the strongest and the primary bonding mechanism for CS-ed particle. Chen et al. [114] also proposed a low-velocity impact-induced metal bonding, in which the conventionally accepted metal jetting and melting may not be prerequisites for solid-state impact-induced bonding.

### 2.1. Cold Spray Parameters

The properties of CSAM-ed parts, such as density, porosity, adhesion, or hardness, depend on the CS spraying parameters, which have to be set to spray the particles in a specific velocity range or deposition window [79,80,85,103]. A velocity of a particle below a $V_{critical}$ or $V_{cr}$ value does not promote the particle bonding, and an excessive velocity,

$V_{erosion}$ or $V_{er}$, results in the erosion of the substrate instead of a deposition consolidation. This ideal velocity depends on the particles' properties and substrate materials [80,115,116]. Table 1 lists the $V_{cr}$ and $V_{er}$ for the most CSAM-ed materials.

**Table 1.** Window of deposition for CS.

| Material | $V_{critical}$ [m·s$^{-1}$] | $V_{erosion}$ [m·s$^{-1}$] | Ref. |
|:---:|:---:|:---:|:---:|
| Al | 625 | 1250 | [79,117] |
| 316L | 550 | 1500 | [79,117–119] |
| Cu | 570 | 1000 | [79,120] |
| Ti | 700 | 1750 | [79,117,121] |
| Ti6Al4V | 750 | 2500 | [122] |
| Ni | 570 | | [117,120] |
| Inconel 718 | 600 | 1700 | [123,124] |

Process parameters optimization is based on particular applications and equipment, working gases, substrate and feedstock materials' characteristics, and others. Typically, these parameters include the gas type, temperature, pressure, nozzle geometry, throat size, and deposition robot strategy. In addition, a critical point is the feedstock powder material itself, particle size distribution, shape, and particle attributes, such as oxide skins and mechanical properties, which influence the ability to form a compacted layer [78,80,83–85,99].

For the CSAM, the working or main gas commonly used is $N_2$ or He, or $N_2$/He mixtures, but for LPCS and MPCS, compressed air is a low-cost option also [2,90,125,126]. $N_2$ has a lower cost than He and, due to the high consumption of the working gas, it is the choice for the main gas. For CS using He instead of $N_2$, the particles are propelled with a higher velocity due to He's higher atomic mass [126–130], e.g., CS-ed 316L (particle size 28 μm) with He reaches 750 m·s$^{-1}$, but less than 500 m·s$^{-1}$ with $N_2$ as the working gas [131]. The CS working gas temperature is set up to high values in the CS gun heating chamber, e.g., 1100 °C for spraying 316L [132]; however, after passing through the nozzle, the gas expands, reducing the density and temperature [131,133]. Lee et al. [134] presented a CFD gas flow simulation in which a CS gas heating chamber at 1200 K and 20 bar resulted in less than 800 K in the CS gas jet, but a velocity higher than 1300 m·s$^{-1}$. Considering the heat transfer inertia from the gas to the particle and the short time of exposition, the temperature of the particles is much lower than 800 K, maintaining the sprayed particles below their recrystallization temperature. It influences the properties of the sprayed material, such as the particles' cohesion, adhesion, strength, and others. For example, for Ti coatings, the cohesion measured by TCT (Tubular Coating Tensile) [135] had a linear relation with the gas temperature [121], and higher cohesion corroborates a material with a lower porosity, higher strength, and DE. However, by selecting a high gas temperature, the cold work and hardness in the material are dwindled by partial recovery and recrystallization phenomena [136,137].

The Standoff Distance (SD) is how far the substrate surface is from the gun nozzle exit. This distance has an optimum value, where the velocity of particles reaches the peak, impacting the substrate with the highest energy possible. A relation presented in the literature as a reference for an excellent SD is seven times the gas jet diameter. Further, the pressure reduces drastically [86], e.g., for a 3 mm gas jet diameter, the SD should be 21 mm. Turbulences, the oscillation of the gas jet, and the irregular distribution of the particles impacting the substrate are also seen to increase the SD, which reduces the DE [138], as confirmed experimentally for CS-ed Al, Cu, and Ti [139]. The adherence of CSAM-ed Ti6Al4V on the steel substrate increased by optimizing the SD parameter, reaching the best value of 50 mm without delamination [140], showing that the relation of an SD seven times the gas jet diameter proposed by Kosarev et al. [86] is just a starting point for parameter optimization and not a rule.

The robot path and velocity influence the characteristics and properties of the CS-ed material; the step between the sprayed single tracks has to be optimized to guarantee good adherence and produce a flat and smooth deposit surface because an insufficient overlapping distance results in a wave surface [138]. Therefore, rotating is one of the most applied strategies for CSAM, building up the part by coating a rotating pipe-like substrate, resulting in parts with symmetry, such as the one presented in Figure 2d, after the post-machining process. This strategy promotes the good adhesion and cohesion of particles but limits the geometries feasible to the symmetrical ones. The use of alternate directions, Figure 5b with the CS laden-jet particles in the Z-direction, increased the material's isotropy when compared to the traditional strategy, Figure 5a with the CS laden-jet particles in the Z-direction, for CS-ed Cu thick parts [141]. Compression tests in the X- and Y-direction indicated different crack propagation paths for the bidirectional strategy, revealing that the robot path influences the preferential direction for crack propagation [142]. The robot path also may change the angle of the impact of particles, drastically affecting the DE and material microstructure. For CSAM, the robot path has a crucial function since the part sidewalls grow up and follow an angle, which has to be rectified to the designed and desired inclination. An adequate robot programme can spray on the inclined sidewall with a jet angle that corrects it, improving DE [92,143–145].

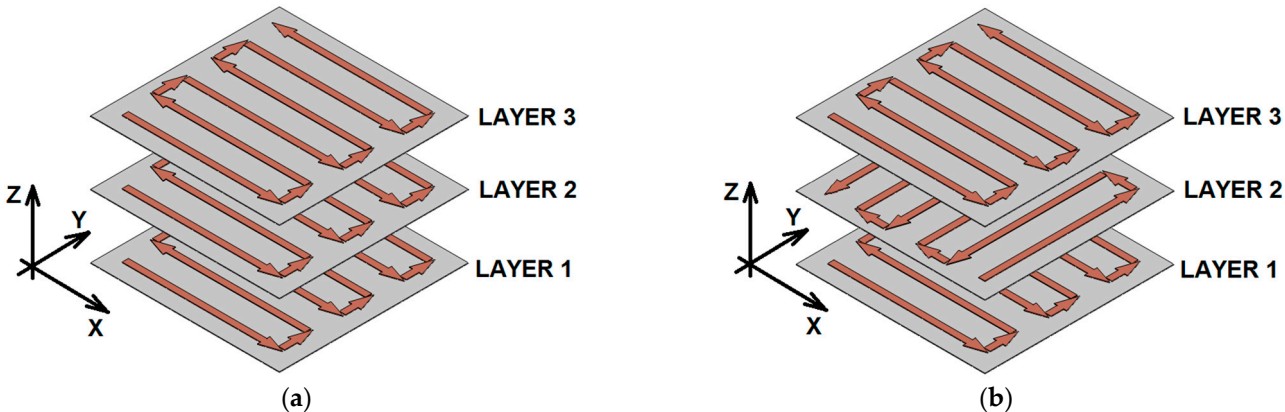

**Figure 5.** Different robot path strategies for CS-ed deposition. (**a**) Traditional or bidirectional, and (**b**) cross-hatching.

CS almost always uses conventional powders as feedstock materials developed for Air Plasma Spray (APS), High-Velocity Oxy-Fuel (HVOF), or laser processes in a spherical and finer particle size range at best. Various techniques are available to produce metallic powders, which are chosen by the chemical composition, characteristics, and/or properties required for the powder [146]. For the CS, the particles' metallurgical, morphological, and physico-chemical characteristics influence the spraying success and material performance [83]. Since CS does not promote recrystallization during the deposition, a deposit with a refined microstructure is obtained by selecting a small grain-size feedstock powder. It improves the mechanical properties; however, a larger gain size promotes more ductility to the particle. Using HT to reach the ideal powder microstructure was an alternative presented by Poirier et al. [147] for CS-ed H13 tool steel and by Story and Brewer [148] for aluminum alloys, resulting in a DE increase from 35 to 60% and from 70 to 90% to Al7075 and Al6061, respectively. Silvello et al. [84] summarized the relationship between powder characteristics, CS process parameters, and the CS-ed material properties by modeling and the experimental results. Table 2 presents coefficients for the model proposed using modeFRONTIER software, in which negative values represent inverse input/output relationships. It is noticed that the particle diameter and hardness influence the CS-ed material characteristics, highlighting the porosity, which is responsible for some CS drawbacks, such as a short fatigue life.

Table 2. Correlation behavior among the different input/output for CS [84].

| Input/ Output | Particle Diameter | Particle Hardness | Gas Pressure | Gas Temperature | Particle Velocity | Deposit Hardness | Porosity | DE | FR |
|---|---|---|---|---|---|---|---|---|---|
| Particle diameter | 1 | 0 | 0 | 0 | −0.431 | −0.187 | −0.213 | 0.104 | 0.097 |
| Particle hardness | 0 | 1 | 0 | 0 | 0 | 0.935 | 0.109 | 0 | −0.324 |
| Gas pressure | 0 | 0 | 1 | 0 | 0.594 | 0.417 | −0.682 | 0.768 | 0.804 |
| Gas temperature | 0 | 0 | 0 | 1 | 0.498 | 0.297 | −0.471 | 0.592 | 0.897 |
| Particle velocity | −0.431 | 0 | 0.594 | 0.498 | 1 | 0.682 | −0.734 | 0.803 | 0.817 |
| Deposit hardness | −0.187 | 0.935 | 0.417 | 0.297 | 0.682 | 1 | 0 | 0 | −0.352 |
| Porosity | −0.213 | 0.109 | −0.682 | −0.471 | −0.734 | 0 | 1 | 0 | −0.819 |
| DE | 0.104 | 0 | 0.768 | 0.592 | 0.803 | 0 | 0 | 1 | 0 |
| FR | 0.097 | −0.324 | 0.804 | 0.897 | 0.817 | −0.352 | −0.819 | 0 | 1 |

CS powders must be characterized before spraying, measuring their particle size distribution by the ASTM B214 standard [149], a sieving separation of the larger and smaller particles, or the laser scattering, classifying the particle size distribution by measuring the laser-illuminated flowing particles. The powder flowability is measured by the time elapsed to flow a certain powder mass through a certified Hall flowmeter, following the ASTM B213-20 standard [150], which is used to measure the powder's apparent density, as indicated by the ASTM B212-21 standard [151]. A previous characterization of the powder is imperative since powders with a flowrate higher than $1\ g{\cdot}s^{-1}$ tend to build up and block the gas flow in the nozzles for LPCS [146]. For HPCS, Vaz et al. [132] presented the flowability for different 316L, resulting in 9 and $17\ g{\cdot}s^{-1}$ for the irregular and spherical shapes, respectively. This powder characteristic impacted the CS powder feeding, which was 0.43 and $0.55\ g{\cdot}s^{-1}$ for the irregular and spherical shapes, respectively. By machine learning, Valente et al. [152] show how to predict a novel powder flowability on a per-particle basis, which can help scholars develop their alloys and powders for CSAM.

An irregular shape of the particles does not necessarily result in a coating or CSAM-ed part with worse properties [153–155]. The high deformation of the CS-ed particles at the impact can act as compensation for their shape irregularity and even for the particle size distribution, which enables using coarse particles, as presented by Singh et al. [153], who obtained similar material strength by coarse and fine Cu particles. CS-ed 316L coatings using water-atomized powders, which had an irregular shape, presented corrosion behavior and a wear-resistance performance very similar to, or even better than, the coatings obtained with spherical gas-atomized powders [132], indicating the viability of using a lower-cost raw material for CS, since the 316L gas-atomized powders are more expensive than the water-atomized ones. Wong et al. [155] obtained very similar porosity values (3.0 ± 0.5%), DE (100%), and hardness (200 ± 10 HV) for CS-ed Ti coatings employing irregular and spherical shape powders, but considering coating quality, the authors suggested the medium-sized spherical powder the best CS option. For Ti6Al4V, spherical particles presented a higher hardness and cohesive strength than a very irregular powder obtained by the Armstrong process, as shown by Munagala et al. [156]. In addition, the powder size distribution influences the CS-ed particles' velocity; smaller particles reach higher velocities than bigger and weightier ones, as presented in a simulation performed for 5, 25, and 50 µm Al particles. The first one resulted in a velocity higher than $600\ m{\cdot}s^{-1}$, but the last one was lower than $500\ m{\cdot}s^{-1}$ [157]. For CS-ed Cu particles, small particles, 5 µm,

reached a velocity of 700 m·s$^{-1}$, while big particles, 90 μm, accelerated up to 300 m·s$^{-1}$. Bagherifard et al. [119] presented a 316L fine powder, $-29 + 12$ μm, with a higher spraying velocity than coarse particles, $-45 + 19$ μm, which resulted in a material with higher particle deformation, mechanical properties, and electrical conductivity. Meanwhile, the $V_{cr}$ is dependent on the particle size, and smaller particles have a much higher $V_{cr}$ than the bigger ones, resulting in an even higher velocity, meaning small particles may not bond, and an optimum size range is achieved for each material, which is generally between 10 and 60 μm. When improving the temperature of particles, $V_{cr}$ is reduced, revealing the need to improve the CS working gas temperature to increase the temperature of smaller particles and the velocity of bigger particles [133,158,159]; however, higher gas temperatures put the equipment in an undesired condition, overloading it and promoting nozzle clogging.

The literature explains how the CS nozzle wall at a high temperature induces clogging because low-melting-point hot particles flow through the nozzle and collide against the nozzle's inner hot wall, inducing the bonding between the particles and nozzle wall, resulting in nozzle clogging [157,160]. Different solutions have been evaluated by researchers aiming to reduce the clogging and improve the nozzles' service life: the assembly of cooling systems surrounding the nozzle to reduce its temperature [157]; redesigning the nozzle for a bi-material component, using glass and WC [161]; aligning the sprayed particles by an electric field and avoiding them to touch the nozzle's hot wall [162]; and others. Clogging can be solved by cleaning methods, such as spraying hard particles at high temperatures or a chemical cleaning with acids. However, besides the monetary loss of clogging, it reduces the DE, can overload the gun chamber dangerously, and imposes maintenance stops during the deposition, generating undesired temperature transitions for large CSAM-ed parts. Sun et al. [10] comment that clogging has been one of the limitations of a more industrial CSAM application.

### 2.2. Challenges for CSAM

CSAM is a technique with great benefits compared to other AM methods. Therefore, it has excellent potential to be implemented in the solid-state AM industry to produce free-standing parts or repair worn components [2,163]. Yet, CSAM is still an emerging technology facing several challenges that need to be studied, such as low as-sprayed geometric tolerances, inferior mechanical properties compared to wrought materials, residual stresses, and low DE-depositing hard materials. In this section, these challenges are discussed, along with the strategies studied to overcome them. Figure 6 presents a scheme of the pros and cons of CSAM over other metal AM processes. It also indicates the advances studied and investigated to overcome the drawbacks.

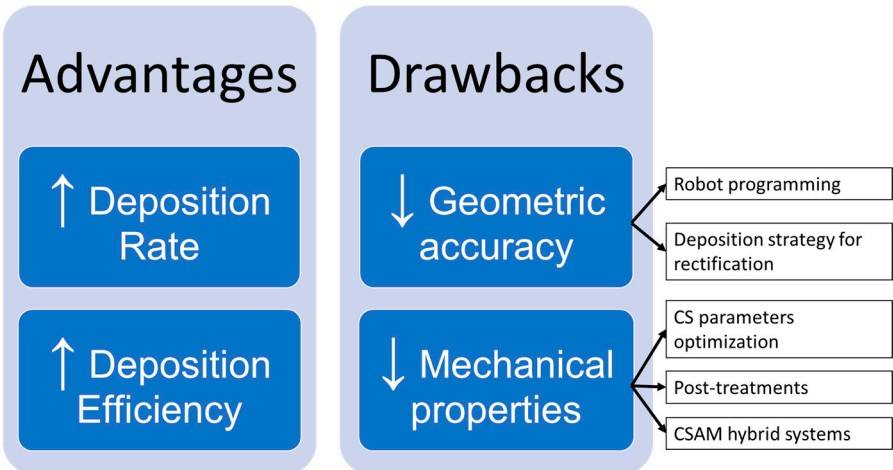

**Figure 6.** CSAM advantages and drawbacks, and the alternatives presented in the literature to overcome them.

### 2.2.1. Possible CSAM Geometries

The AM technologies rely on building an object layer by layer. Thus, controlling geometric tolerances is imperative to produce complex shapes or near-net-shape parts. Still, CSAM has to be more precise, and the literature presents some reasons for this CSAM limitation. First, because the velocity profile of the particles in the jet spot that exits the nozzle is uneven, the center of the laden-jet particles has a higher density of particles and greater velocities, promoting a superior deposition on this region than on the jet periphery. Cai et al. [138] simulated the single-track deposit profile, concluding that a 2D distribution profile approximately fits a Gaussian curve. Ikeuchi et al. [164] evaluated different machine learning approaches to accurately preview the CSAM track profile, saving much experimental time and CSAM spraying costs. Furthermore, Kotoban et al. [165] investigated the relationship between the shape of a single-track coating and the DE, concluding that in the first layer deposition, the particles on the jet periphery have a slight decrease in DE compared to the jet core, producing a triangle shape deposit that sharpens layer by layer. Finally, Wu et al. [166] developed a model to compensate the layer thickness by optimizing the robot velocity at the different regions on the substrate surface, resulting in a smoother CS-ed material surface.

Knowing that CSAM-ed deposits tend to produce pyramid-shaped coatings, some robot path trajectories and strategies have been developed to obtain near-net-shape parts [144,167–173]. For instance, Wu et al. [167] established a new stable layer-by-layer building strategy that sprays at a deflected angle towards the inclined walls of the pyramid-shaped coating, which allows building components with straight walls. Another example is the work of Vaz et al. [144], where a new method was implemented that consists of spraying with a circular movement at an angle different than the normal and allows free-standing building with controlled shapes, as presented schematically in Figure 7, but well described in the literature by Vaz et al. [144]. Yet, further studies on deposition strategies and the production of free-standing components are encouraged since they can expand the application areas of CSAM.

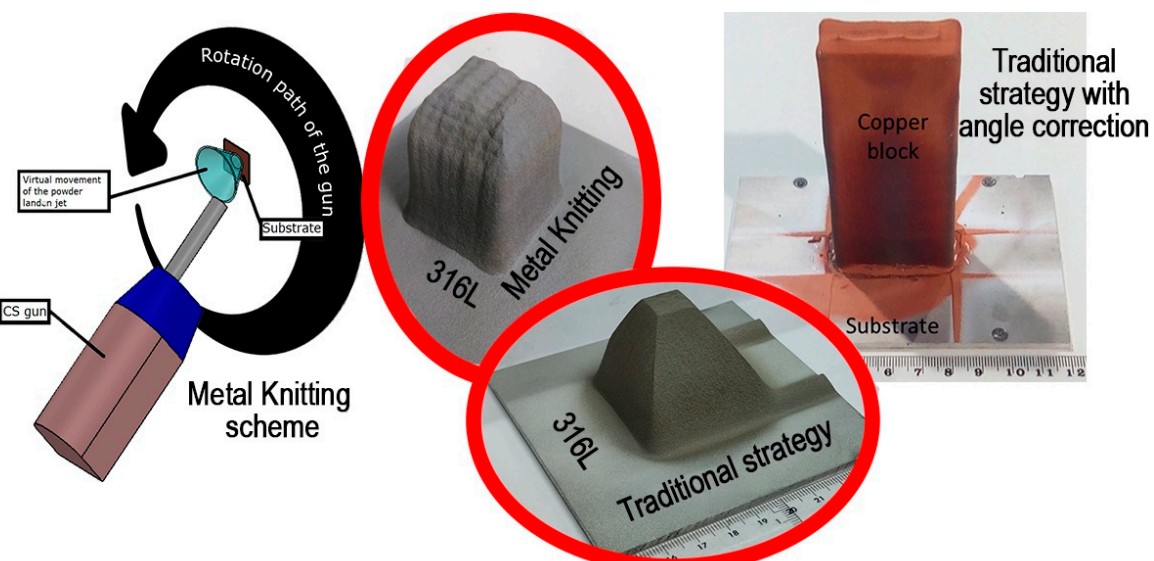

**Figure 7.** Metal Knitting, a CSAM alternative strategy to the traditional deposition. Reprinted with permission from Ref. [171], Springer Nature, 2020. Unit: mm.

CSAM can produce arrayed structural components, as presented in Figure 8 for CSAM-ed Ti on stainless steel. It was made by masking the substrate to shadow the areas where the sprayed material was not supposed to cover. Masking has been presented in the literature for other thermal spray processes, using tapes, pastes, shields, or other high-temperature resistant material removed after the coating deposition [174–177]. CFD has been developed

to understand the influence of the masks on the CS gas flow, disturbances on the particle's trajectory, and the formation of bow shockwaves, which reduces the gas velocity [178]. It suggests using a higher particle velocity and setting the CS parameters to suppress this harmful effect of the masking strategy. Klinkov et al. [179] presented a model showing the impact of the mask on the particle behavior, velocity, and trajectory. The distance of the mask to the substrate cannot be excessive because it affects the deposition geometry, decreasing the width of the masked zone and diminishing the accuracy of the CSAM-ed geometry. An industrial application of the CSAM masking strategy is the fabrication of compact heat exchangers for electronic devices [180–183]. As well as array structures, diverse geometries are feasible, such as Braille impression for blind people or raised areas in molds for plastic injection, among others.

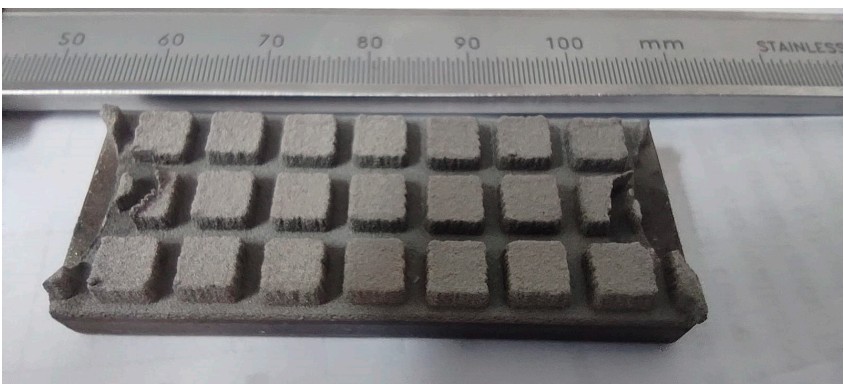

**Figure 8.** CSAM-ed Ti on stainless steel using masking strategy. Unit: mm.

Applying CSAM with other processes is a hot topic for industrial applications, e.g., for a unique component, some regions can be CSAM-produced, which is faster, and others can be DMLM-made, resulting in more details or complex geometries. Another CSAM use is joining dissimilar materials because CS has no metallurgic union with the substrate. It is helpful for composites, e.g., a carbon-fiber-reinforced polymer or a sintered SiC. This Al interlayer strategy was tested by Xie et al. [184] for joining TiN/Ti6Al4V, but using a hot dip to make a 25 μm-thick Al coating; however, it can promote an undesired atomic diffusion depending on the materials and process temperatures and time, which is prevented by using CSAM. These hard- or impossible-to-weld materials can have a surface coated by a thick CSAM weldable material, e.g., Al or 316L, which can be joined on other structures quickly. Champagne Jr and Champagne III [185] presented this method for using CS directly as the joining element and growing a CSAM-ed volume on the part to be arc-welded on another element. This joining was tested for a light-alloy magnesium ZE41A-T5, employing Al as the filler metal [186], joining Al 6061 to a ZE41A Mg alloy using CS sprayed Al as the transition material on the Mg alloy surface and welded to Al 6061 by FSW [187,188], and joining Al to Cu by FSW with a Ni interlayer [189]. Daroonpavar et al. [190] presented CS with the capability to make a corrosion-resistant coating on an AZ31B Mg alloy, employing the metallurgically incompatible Ta–Ti–Al layers, also reducing the wear rate from $10^{10}$ to $10^8$ $\mu m^3 \cdot N^{-1} \cdot m^{-1}$.

### 2.2.2. Improving the Mechanical Properties

To date, CS processes have been used mainly in the aerospace, automotive, marine, and defense industries, where the performance requirements of the deposits are very demanding [99,163]. Therefore, one of the main issues with CSAM is the mechanical properties of the deposits. Apart from the hardness, which tends to be greater than the bulk due to the cold work hardening of particles during impact [132,144,191], the as-sprayed deposits present less favorable mechanical properties, such as lower strength, ductility, electrical and thermal conductivity, and wear resistance. It is attributed to the inherent microstructural defects of the CS process, such as micro-pores and interparticular

boundaries [2,191]. Moreover, as the particles are arranged layer by layer, anisotropic responses have been reported in the literature for CSAM-ed deposits [141]. The literature presents anisotropy for other processes that deform the material in a preferential direction, e.g., cold rolling [192–194], extrusion [195], friction stir welding [196], or even laser AM processes [197,198]. For CSAM, high isotropy was observed in a plane parallel to the substrate surface [141,199–201], but in a vertical or Z-direction, the material had lower strength. This behavior is presented in the literature for CSAM-ed Cu [171], Al [202], and 316L [203].

Moreover, the use of CS is also limited by the intrinsic characteristics of the materials. For example, only soft and ductile materials, such as Cu and Al, are easily deposited, which is deducted from the number of papers linking "cold spraying" and "aluminum" or "copper" keywords. In contrast, hard materials (e.g., Maraging steels, Ti6Al4V, Inconel 718, etc.) with the poor capability to deform at a solid state can hinder the formation of a dense component [28]. Therefore, recent studies focus on optimizing the CS process parameters to obtain the ideal $V_{cr}$ for each material so that quality coatings are produced [77,99,123,204–206]. For instance, Li et al. [205] did a literature review on the solid-state CS-ed Ti alloys, focusing on the process parameters, deposition characteristics, and limitations of these materials. Another example is the work of Pérez-Andrade et al. [123], which presents the optimization of parameters and post-treatment processes for obtaining high-quality thick deposits of Inconel 718 for AM applications.

CS-ed coatings also tend to be influenced by compressive residual stresses generated by the severe impact deformation of the particles. Such compressive stresses can be beneficial up to a certain point. However, if they are too high, the adhesion of the deposit is usually hindered, and a crack can nuclei and grow in the interface substrate/coating, or it can completely detach, de-coating from the substrate [207,208]. For CSAM, these residual stresses are accumulated layer by layer, and if the particles have poorly adhered to the substrate, the deposit separates from the substrate. Making freeform parts is not a problem because the substrate has to be eliminated and only acts as a base or support. Still, with the employment of CSAM as a repairing method, this detachment and low adhesion is highly prejudicial of the excellent performance of the repairing service.

These challenges represent a drawback for CSAM compared to other AM methods. Nevertheless, several process strategies have been successfully explored in the literature, such as post-processing methods (e.g., HT) or hybrid deposition technologies, such as Laser-Assisted Cold Spray (LACS) and Cold Spray Shot Peening (CS-SP).

Heat Treatments (HT) are one of the most effective ways to enhance the microstructure of CSAM-ed deposits [2]. The tailoring of the final properties of a broad range of materials with HT, such as Cu [136,208–210], Al alloys [208,211–213], Ti alloys [208,214–216], Ni alloys [81,123,124,204,217–219], 316L [119,208,220], among others, is reported in the literature. Furthermore, HT relieves residual stresses, reduces the microstructural defects (e.g., porosity and particle boundaries), and improves the cohesion between particles which significantly influences the material performance, since the failure mechanism during the stress loading changes from an interparticular mode to a cleavage-like and ductile mode. In the first one, the crack grows surrounding the particles and detaches one to the other. HT promoted a metallurgical bonding of particles, increasing cohesion, material strength, and plasticity or ductility. Dimples evidenced it in SEM images of the fracture surface [202,210,221–223]. For Inconel 718, Sun et al. [124] applied induction for heating the material, which represents a possibility to select the CSAM-ed part region to be HT-ed, instead of the whole material, e.g., treat only the component areas that are exposed to wear or friction. The induction HT promoted the cohesion of the particles by the eddy current, as well as the atomic diffusion, which resulted in higher mechanical properties due to higher dislocations and twin densities in the neck formed between the particles than in the particles' center [224]. Due to the hardness reduction, Zhang et al. [225] presented the HT-positive effects on the post-machining process of Al7075. Another heating process is Electric Pulse Processing (EPP), in which applying high-density electron charges through

the material promotes changes in the microstructure and mechanical behavior of alloys, such as precipitates distribution, yield strength, elongation, and hardness [226]. For example, for CSAM-ed Cu, Li et al. [210] show an expressive improvement in its mechanical properties, reaching a UTS of 200 MPa over 100 MPa in the as-sprayed condition and elongation of 20% over 2%.

Particularly, annealing is considered a simple post-processing method that positively impacts the as-sprayed CSAM microstructure. It promotes diffusion and recrystallization processes that mitigate the undesired microstructural defects and change the mechanical properties; the work-hardened deposits are softened, increasing their ductility, but reducing their hardness compared to their as-sprayed counterparts.

Spark Plasma Sintering (SPS) is a technique developed for ceramics and powder metallurgy that has improved CSAM-ed density and mechanical properties. SPS is pressing compacted powder and applying a pulsed current discharge that can reach thousands of Amperes but low voltage under pressure. It generates plasma between the intimate close particles, which results in micro welding, forming necks at contact points, atomic diffusion, and plastic flow [224,227–229]. In addition, Joule heating and plastic deformation enhance the sinter's densification, improving the particles' cohesion and material strength [230]. For the CS-ed TiC–Cu composite, SPS eliminated the interparticular region [231]. The SPS temperature was directly related to improving the mechanical properties, ductility, and decreasing the hardness of the CSAM-ed Cu, as presented by Ito and Ogawa [230], who selected 50% of the Cu melting point as the maximum SPS temperature for 5 min. This short time is an advantage of SPS over annealing, which typically keeps the material in the furnace for hours. Figure 9 shows the microstructures of CSAM-ed Ti6Al4V as-sprayed and after SPS post-treatment.

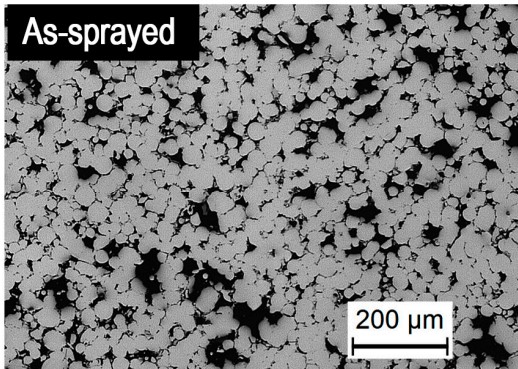 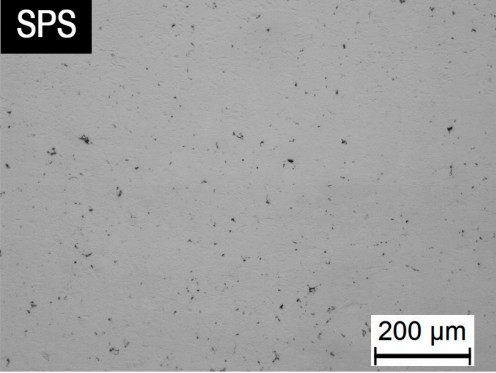

**Figure 9.** SPS effect on CSAM-ed Ti6Al4V microstructure.

Another HT that has recently drawn attention in the field of CSAM is Hot Isostatic Pressing (HIP), which is presented in detail by Bocanegra-Bernal [232] and by Atkinson and Davies [233]. The HIP technique can be used directly to consolidate a powder or supplementary to densify a cold-pressed, sintered, or cast part. This method can eliminate the pores and micro-cracks of the material by compressing the samples with high temperatures, e.g., 1000 °C for Ti alloys, to an isostatic pressure in the order of hundreds of MPa at the same time, resulting in fully isotropic material properties [234,235]. It has been successfully applied to metals, composites, and ceramics obtained by different processes. However, few studies are available in the literature for CSAM, and they are focused on hard materials that are difficult to deposit by CS, such as Ti [236,237], Ti6Al4V [237–239], and Inconel 718 [123].

Figure 10 presents the densification and phase changes, precipitating β in an α matrix, in CSAM-ed Ti6Al4V employing $N_2$ and He as the CS working gas. However, this post-treatment cannot close exposed porosity because the HIP gas fills these pores. A solution is a pre-HIP process of encapsulating the sample and converting those into internal pores to be removed by the HIP. The HIP also cannot remove large internal pores since diffusion bonding does not occur when metal/metal contact is not intimate. It happens when the

CS-ed material has low plasticity even in high temperatures, if the surfaces of the internal defect are oxidized, or if there is a gas inside the pore that does not diffuse, e.g., air, He, or $N_2$ [232]. It represents a limitation for CSAM HIP use if the CS-ed deposition process cannot produce parts with very low porosity and a very thin interparticular region, which occurs when spraying low-ductility powders.

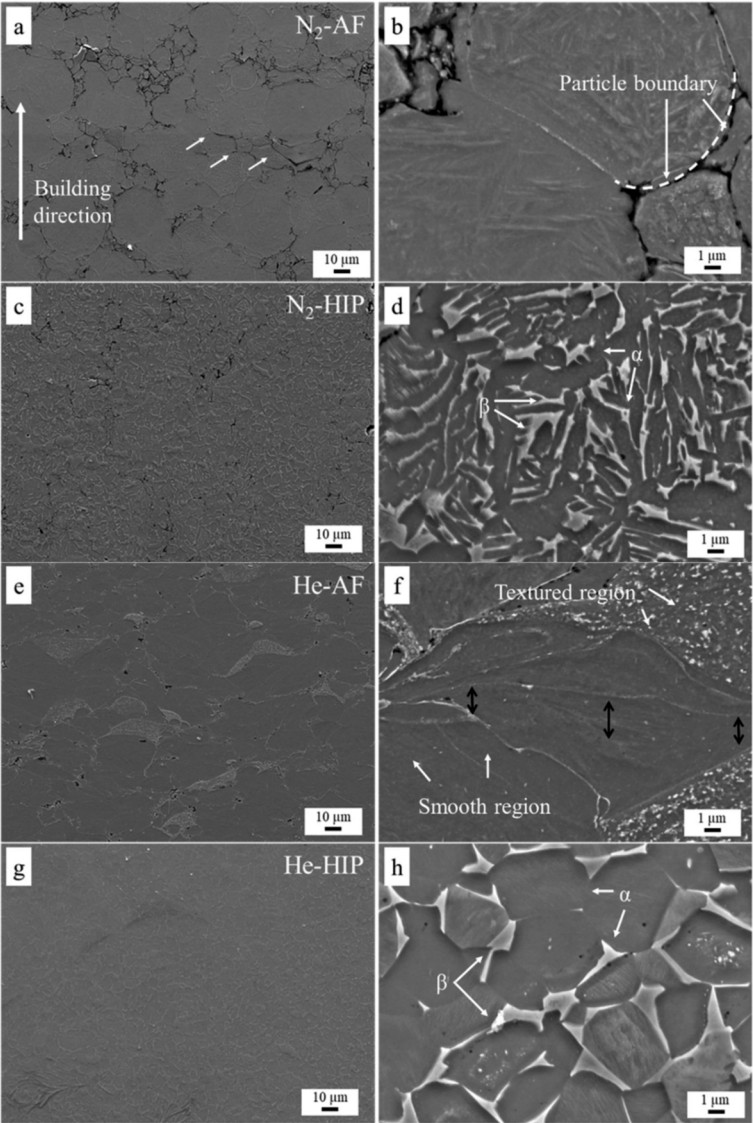

**Figure 10.** Effect of HIP on the microstructure of CSAM-ed Ti6Al4V, densification, and phase change. SEM images with (**a,c,e,g**) low magnification and (**b,d,f,h**) high magnification. Reprinted with permission from Ref. [238], Elsevier, 2019.

The melting or sintering AM processes drastically change the microstructure of the feedstock material during the processing, and CS represents an advantage over SLM or DMLM in this point. However, a hybrid process of coating a CSAM-ed part can significantly improve its wear and corrosion performance, as presented by Vaz et al. [240] coating CSAM-ed Maraging with HVOF-sprayed WC. In addition, Feng et al. [241] used induction heating to remelt AlCoCrCuFeNi HEA, improving the wear resistance by phase transformations. Laser remelting or glazing has been investigated as a post-treatment on CSAM, eliminating micropores within the deposit and enhancing the cohesion of particles. Remelting changes the ASI and other CS bonding mechanisms for metallurgical bonding. Laser remelting of CS-ed Al onto steel substrate presented an FeAl intermetallic formation, improving its wear resistance [242]. For Ti, Astarita et al. [243] and Marrocco et al. [244] obtained a



thin and dense remelted layer, which improved the corrosion behavior in a 3.5% NaCl solution, reaching the same performance as a wrought Ti bulk. Kumar et al. [245] showed an improvement in the wear resistance for the Ti-based MMC.

The laser glazing applied on CS-ed Inconel 625 eliminated the cold-worked microstructure, generating a columnar dendritic one. It reduced the hardness but eliminated the interconnections between the pores, increasing the material corrosion performance [246]. Zybala et al. [247] improved the surface hydrophobicity of CSAM-ed irregular powders Ti6Al4V and Ti after laser surface post-treatment. This condition is attractive for CSAM-ed parts developed for the Oil and Gas sector, where Ni-based alloys have been employed as $CO_2$ and $H_2S$ corrosion-resistant materials. Other cheaper heat sources can be studied, and it is a lack in the literature, e.g., PAW, used by Pukasiewicz et al. [248,249] for HVOF-sprayed FeMnCrSi coatings, or GTAW, applied by Zabihi and Soltani [250] for FS Al-based MMC coatings.

Plasma Electrolytic Oxidation (PEO) or micro-arc oxidation has been used to produce hard ceramic coatings on Al and other alloys [251]. It improves their wear and corrosion resistance due to the formation of the protective ceramic coating on the material enabled by the plasma discharges, supported by an aqueous electrolyte [252]. For CS-ed Al, Rao et al. [253] presented a stable, well-adhered, and harder PEO layer formed on the CS-ed Al7075 coating, 1353 and 144 HV, respectively, which resulted in a higher dry sliding wear resistance. It also improved the corrosion resistance, with a three-order lowered corrosion current density. Using PEO on CS-ed Al + $Al_2O_3$ on a Mg alloy substrate, Rao et al. [254] reduced the sliding and abrasion wear rates ten times, mainly due to the increase in hardness from 700 to 1300 HV.

The infrared irradiation as a heating source for the HT of CSAM-ed Cu alloys was tested by Chavan et al. [255]. This heat source is cheaper than laser equipment and has a wavelength range similar to lasers. Still, a significant advantage of infrared irradiation is the absence of a furnace, a chamber, or a controlled atmosphere, as occurs for annealing. It enables this system for CSAM in situ repairs or repairing large components that do not fit in conventional furnaces, e.g., oversized axles or injection molds. Infrared irradiation has been previously used for arc-sprayed Zn alloys coatings, improving wear and corrosion resistance [256].

Shot Peening (SP) is a post-treatment technique of cold working by propelling glass, ceramic, or steel balls against the material, reducing the material's surface roughness, inducing the surface compressive residual stress, and, consequently, increasing the part's fatigue resistance by retarding crack initiation [257]. Moridi et al. [258] showed the SP applied for CSAM-ed Al6082, reduced roughness from $R_a$ 12.4 to 4.7 μm and improved the depth of the compressive stress layer from 350 to 400 μm, but without a significant compressive stress value improvement. Furthermore, due to porosity and plastic deformation reduction, the hardness and corrosion resistance were improved in CS-ed Zn by SP post-process [259]. Similar mechanisms and results were obtained by Ball-Burnishing (BB), a process in a ceramic or hard ball, with a diameter of <10 mm, which compresses and deforms the CS-ed material, as occurs with SP, but without impact, more similar to a rolling process. BB was applied to CS-ed 17-4PH stainless steel, improving the depth from 130 to 190 μm of 200 MPa compressive residual stress [260,261]. In general, any technique that improves the part's fatigue life is attractive for CSAM; however, SP has presented low effectiveness and does not indicate more research interest or industrial promisor use.

Waterjet Cavitation Impact (WCI) is another technique presented in the literature that should be tested for CSAM, since there is a lack of this in the literature, aiming to improve the material surface properties, especially the compressive residual stress. Cavitation is a phenomenon in which the static pressure of a liquid reduces to below the liquid's vapour pressure, forming microbubbles that collapse when subjected to a higher pressure. It generates shock waves that impact the material surface [262]. For WCI, the material is exposed to a water jet under controlled conditions, promoting its plastic deformation and densification, as occurs for SP. WCI has been applied in the industry since the 1990s for

parts produced by different techniques and exposed to fatigue degradation, such as gears, shafts, and other [263]. Good results have been exposed in the literature, Zhang et al. [264] reached 175 MPa compressive residual stress in a 2A12 Al alloy by a WCI with a water jet under 75 MPa and 20 degrees off-normal inclined; Soyama and Okura [265] presented how WCI resulted in a significant improvement in the fatigue life of Ti6Al4V.

Cold Rolling (CR) was experimented with for CS-ed Cu on steel by Bobzin et al. [266], resulting, after a 14% thickness reduction, in cracks and delamination on the coating/substrate interface, which was resolved by annealing at 500 °C before CR, a Thermo-Mechanical Treatment (TMT) or Hot Rolling (HR). It resulted in good adhesion to the substrate without cracks in the Cu coating but lower hardness due to the annealing process that dwindled the cold working in the particles from the CS-ed deposition. Tariq et al. [267] employed TMT for CSAM-ed Al-$B_4$C MMC, reducing by 60% in thickness, improving the mechanical properties, increasing UTS from 35 to 131 MPa, and elongation from 0.5% to 5% due to the interfaces of the particles dramatically enhanced by the diffusion activity. TMT has been applied for materials that have poor formability at room temperatures, such as Mg alloys [268] or TiAl-based alloys [269,270], as well as the CSAM-ed Ni-Al [271], A380 alloy [272], and Al-$B_4$C [267]. Depending on the CSAM part's geometry designed, such as plate-like, TMT is adequate. However, TMT promotes a considerable anisotropy for a bulk-like shape due to the unidirectional plastic deformation induced by the post-treatment.

Friction Stir Processing (FSP) imposes a friction force on the CSAM-ed material that softens the surface, increasing the amount of shear straining in the processed region and promoting dynamic recrystallization. Microstructural changes and grain refinement showed this, altering the CSAM-ed material's mechanical properties, porosity, and cohesion of particles [273]. The literature presents the FSP applied for Al alloys [274], Mg alloys [275], MMC [276], and 316L [220], focusing on the improvement of their tribological and corrosion performance. Ralls et al. [220] studied CS-ed 316L + HT + FSP, showing that the post-treatments eliminated the δ-ferrite contained in the powder by atomic diffusion. In addition to that, the authors observed a reduction in porosity to values close to zero and hardness from 330 to 190 HV, as HT-ed; however, FSP increased it from 190 to 280 HV. On the other hand, FSP harmed the CSAM-ed 316L wear resistance, from $2.27 \times 10^{-9}$ to $1.02 \times 10^{-9}$ mm$^3\cdot$N$^{-1}\cdot$mm$^{-1}$. Table 3 summarizes the CSAM post-treatments presented in the literature, considering their main effects and results studied by scholars.

**Table 3.** Post-treatments applied for CSAM.

| Material | Post-Treatment | Post-Treatment Effects Obtained | Ref. |
|---|---|---|---|
| Cu | HT | Improved conductivity, mechanical properties, isotropy, and ductility; reduced hardness. | [208,210,230,277,278] |
| Cu | SPS | Improved mechanical properties and ductility, reduced hardness. | [230] |
| Cu | FSP | Microstructure changed, refining grain size, improved mechanical properties and ductility, reduced hardness | [210] |
| Cu | EPP | Microstructure changed, refining grain size, improved mechanical properties and ductility, reduced hardness | [210] |
| Cu-Al | Infrared irradiation HT | Improved electrical conductivity, maintained the elastic moduli, improved cohesion of particles, reduced hardness. | [255] |
| TiC-Cu | SPS | Promoted phase change and sintering Ti-C-Cu, eliminated interparticular region, increased hardness. | [231] |
| Al6082 | SP | Improved compressive stress layer depth, changed the fatigue fracture mechanism from intercrystalline to transcrystalline. | [258] |
| Al-Mg-Sc-Zr | HIP | Maintained a very low porosity, improved mechanical properties, improved the compression resistance. | [237] |

**Table 3.** *Cont.*

| Material | Post-Treatment | Post-Treatment Effects Obtained | Ref. |
|---|---|---|---|
| Al-Al$_2$O$_3$ | HT | Promoted phase change, reduced porosity and hardness, improved mechanical properties and ductility. | [213] |
| Al-B$_4$C | HT | Improved mechanical properties and ductility, reduced hardness. | [267] |
| Al-B$_4$C- | TMT | Improved adhesion, mechanical properties, and ductility, reduced hardness. | [267] |
| 316L | HT | Reduced porosity and hardness, maintained phase composition, improved ductility and fatigue performance. | [119,208,220,279] |
| 316L | HIP | Reduced porosity and hardness, maintained phase composition, improved ductility and fatigue performance. | [119,237] |
| 316L | HT + FSP | Improved mechanical properties, reduced porosity, reduced hardness negligibly, reduced the wear resistance. | [220] |
| Ti | HT | Maintained the same porosity, increased the mechanical properties and ductility. | [208,216,280] |
| Ti | HIP | Reduced porosity from 4.3 to 2.2%, improved mechanical properties, changed pores morphology. | [236,237] |
| Ti | Remelting | Reduced hardness, transformed microstructure, eliminated interparticular region, improved corrosion behavior. | [243,244,281] |
| Ti6Al4V | HT | Reduced porosity, promoted phase changes, improved mechanical properties and ductility, reduced hardness, | [154,214,282] |
| Ti6Al4V | HIP | Reduced porosity, promoted phase changes, grain refine, improve mechanical, improve the ductility. | [237–239,282] |
| Ti6Al4V | Remelting | Improved hardness, increased surface roughness, coefficient of friction in wear testing, and tensile residual stress. | [247,283] |
| Invar 36 | HT | Improved mechanical properties, ductility, reduced the compressive residual stress. | [284] |
| Inconel 625 | HT | Increased hardness and the fatigue performance. | [217] |
| Inconel 625 | Remelting | Reduced hardness, transformed cold worked microstructure in the particles to columnar dendritic, improved corrosion behavior. | [246] |
| Inconel 718 | HT | Reduced porosity, improved mechanical properties and ductility, reduced the compressive residual stress. | [81,124,204,218,219, 285] |
| Inconel 718 | HIP + solution HT + aging HT | Reduced porosity and compressive residual stress, improved conductivity. | [123] |

### 2.2.3. Avoiding Post-Treatments

The first post-treatment needed for CSAM-ed parts is the machine processes because CSAM cannot produce parts with the final geometry or roughness, which is a challenge for CSAM, as stated by Kumar and Pandey [126]. However, with the development of more complex robot manipulations, the machining has been planned for specific and essential areas of the component, such as bearing houses, screws, or axles journals, among others.

Regarding the materials' properties and characteristics, Laser-assisted Cold Spray (LACS), also called Supersonic Laser Deposition (SLD), is a relatively recent manufacturing process that combines the CS process with a complementary laser that heats the deposition zone while spraying. This method combines the benefits of both technologies, the CS solid-state deposition of metals at short times with little material waste and the bonding strength by heating the deposition zone with a laser without increasing the oxygen levels within the deposit [108,286,287], even applied for the LPCS process [288]. Lupoi et al. [289] and Bray et al. [290] presented LACS as an option to suppress the disadvantage of N$_2$

as a working gas (with a low particle velocity) by the implementation of a laser source to illuminate the spraying location. It softens the substrate during the CS deposition, promoting particle plasticity, even by phase transformation. Barton et al. [291] showed an Fe-based alloy transforming the ferritic into an austenitic phase, which is more ductile; and Birt et al. [287] concluded that LCAS is capable of depositing Ti6Al4V using $N_2$ as the working gas with densities as high as or higher than those deposited using He without the laser assistance. Furthermore, the adhesion of CoNiCrAlY onto Inconel 625 and Cu onto Al were improved by particles/substrate local melting at a micro level and intermetallic formation [292], heating the particles to 80% of the powder melting point [293].

Another LACS option presented in the literature is using the laser to heat, clean, and ablate the substrate milliseconds before the CS deposition. It intends to soften the substrate, allowing the particles to deform and consolidate CS-ed material at an impact velocity lower than its $V_{cr}$ [294]. The use of CSAM for hard materials has been one of the biggest challenges for the industry and researchers.

Regarding the selection of laser parameters, the laser ($CO_2$, Nd-YAG, or Yb-fiber), wavelength, pulse duration, and energy affect the penetration depth of the thermal energy transferred by phonons to the metal [295]. Therefore, their optimization and selection depend on the materials' characteristics and properties. For example, some authors presented experiments for a LACS employing power between 1 and 5 kW, with expressive benefits to the CSAM microstructure and DE using high-power laser assisting [293,296–298]. Still, an excessive heat input can result in grain growth and hardness reduction, i.e., the LACS process results in annealing effects on the cold-worked particles during deposition [299].

Overall, LACS increases the temperature of the particles at the impact, improving the DE and reducing porosity in the deposit microstructure, as presented by Olakanmi et al. [296], reaching pore- and crack-free Al-12Si CS-ed on stainless steel. LACS also broadens the range of CS-ed materials [290]. As a result, LACS has successfully deposited dense parts of hard materials with high DE, such as oxide-free Ti [290], Ti6Al4V [287], MMC [300], Stellite-6 [289], CrMnCoFeNi high entropy alloy (HEA) [301], $Fe_{91}Ni_8Zr_1$ [291,299], 15-5 PH stainless steel [302], and AISI 4340 [297]. Another methodology employed for substrate pre-heating and adhesion improvement was induction, presented by Ortiz-Fernandez and Jodoin [303], spraying Al onto Ti6Al4V, resulting in higher DE and lower porosity.

During CS, particles are accelerated and sprayed at high velocities. At the moment of impact of the first layer, these particles are deformed and remain attached to the substrate. In the subsequent layers, the particles now impact the deposited material, causing the so-called tampering or tamping effect: new particles crush the previous layers of deposited material, causing compaction of the coating, thereby reducing porosity, a peening effect [89,304]. This effect can also be activated by mixing larger particles with the CS feedstock particles to deform the deposited material [305], as presented by Ghelichi et al. [306], mixing −30 + 5 with −90 + 45 μm Al particles, and Lett et al. [140], mixing −45 + 15 with −250 + 90 μm Ti6Al4V particles. Luo et al. [219,307] studied the effects of in situ CS-SP on the microstructure of Ti, Ti6Al4V, and Inconel 718. They presented it as an effective way to increase the DE while reducing porosity and improving inter-particle bonding and cohesion.

However, the −24 + 7 μm Inconel 718 were mixed with −187 + 127 μm stainless steel particles in different concentrations, and this last did not participate in the final sprayed material. Still, it acted for the peening effect, resulting in a drastic porosity reduction from 5.5 to 0.2%, improving the DE from 22 to 33%, and improving the hardness from 420 to 510 $HV_{0.3}$ [219]. For CS-SP-ed Ti6Al4V, increasing the mass of larger particles from 50 to 90% vol. in the feedstock powder, the porosity reduced from 6 to 0.2% [140] and induced compressive residual stress of 444 MPa·m$^{-1}$, instead of the tensile residual stress of 126 MPa·m$^{-1}$ obtained without SP. Hybrid use of CS was shown by Li et al. [308], spraying on the AA2219 alloy GTAW welded joint, altering the residual stress drastically and even promoting the compressive residual stress in some areas near the welding bead. Daroonparvar et al. [305] listed the use of CS-SP for different coatings on Mg alloy

substrates, using other materials for the coating and SP; e.g., Ni with 410 stainless steel 150–200 μm and Al6061 with 1Cr18Ni stainless steel 200–300 μm, among others.

### 2.2.4. Measuring of Properties

Regarding the characterization, CSAM-ed samples have been evaluated as other thermal sprayed coatings. Conventional characterization techniques, such as optical or light microscopy, SEM, and microhardness have been seen in the literature [309]. These are potent tools for experts in the CSAM theme because the researchers can infer important materials' properties from the material microstructure. By image analysis, the Flattening Ratio (FR) is obtained, a measure of the compression of a sphere along a diameter to form an ellipsoid-like splat; the higher the FR, the higher the material plasticity. Electron Back Scattering Diffraction (EBSD) is a technique capable of identifying material phases at each analysis point and presenting the 3D orientation of the crystal lattice at each point. It has been used in CSAM to interpret the orientation in crystallographic planes in the different particles, which is even more important for the characterization of CSAM post-treatments [200]. Figure 11 shows CSAM-ed Ni/FeSiAl in as-sprayed and HT-ed conditions, Figure 11a,b, respectively. HT promoted a recrystallization, grain coarsening, and phase transformation in the material, as interpreted from the size of each colored area, which is more significant in Figure 11b. This image has fewer areas without a defined atomic lattice plane of the crystalline structure, as seen in the as-sprayed condition as dark green. These dark green areas represent patterns uninterpreted by the detector, which is related to the severe particle deformation during the CS deposition. The improvement in this indexing rate from 78 to 90% represents the recrystallization phenomena from HT post-treatment [310].

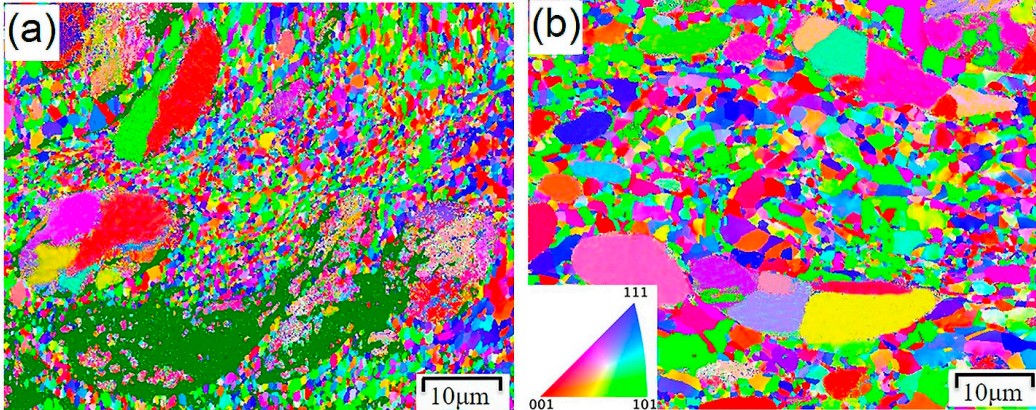

**Figure 11.** EBSD image of CSAM-ed Ni/FeSiAl before and after HT. (**a**) As-sprayed condition and (**b**) annealed condition. Reprinted with permission from Ref. [310], Elsevier, 2019.

Porosity has been calculated by CSAM cross-section image analysis [130,132,311], but other techniques have been presented in the literature as alternatives for higher accuracy in porosity measurements or a non-destructive approach, e.g., gas or He pycnometry [312,313], X-ray microtomography [314,315], laser-ultrasonic inspection [316], water absorption or the Archimedes method [317], and electrochemical impedance spectroscopy [318]. In addition, microhardness (Vickers and Knoop) employing low loading and nano-hardness techniques have been used to determine the hardness gradient in single particles. At the same time, microhardness utilizing higher loadings results in a macro evaluation of the material and fracture toughness by interpreting cracks grown due to the indenter loading [309,319]. Furthermore, the same Berkovich indenter used for the nano-hardness test provides the material elastic moduli, as described in the ISO 14577-1:2015 standard [320–322], an important property to preview the deformation of the material under the service loading.

Using CSAM for repairing processes or as a hybrid stage above a substrate made by other processes infers the need for good adhesion, which is the bonding strength between

the CS-ed material and substrate. For thermally sprayed coatings, the ASTM C633-13 standard [323] is the most used technique to measure its adhesion, known as Tensile Adhesion Testing (TAT), which is basically comprised of a thermal-spray-coated disk, dia. 1 in., that is attached with epoxy to a complimentary uncoated plug and detached by a uniaxial tensile loading, the relation loading-area results in the bonding strength, in MPa or ksi [309,324]. However, for bulks, ASTM C633-13 [323] is inadequate. A technique within the sample machined from the CSAM-ed freeform part has been presented in the literature as a more effective method, modified tensile testing, based on the ASTM E8-22 standard [325]. Ichikawa et al. [326] machined adhesion samples of CSAM-ed Cu onto an Al substrate, eliminating the interference of a bonding agent, epoxy adhesive, and guaranteeing the rupture in the Cu/Al interface. Boruah et al. [327] used a similar technique, but with CS Ti6Al4V on a washer surrounding an exposed pin-like substrate, which are tensile together, rupturing in the coating/substrate interface. Figure 12 shows the schemes for TAT and ASTM E8-22 modified adhesive testing.

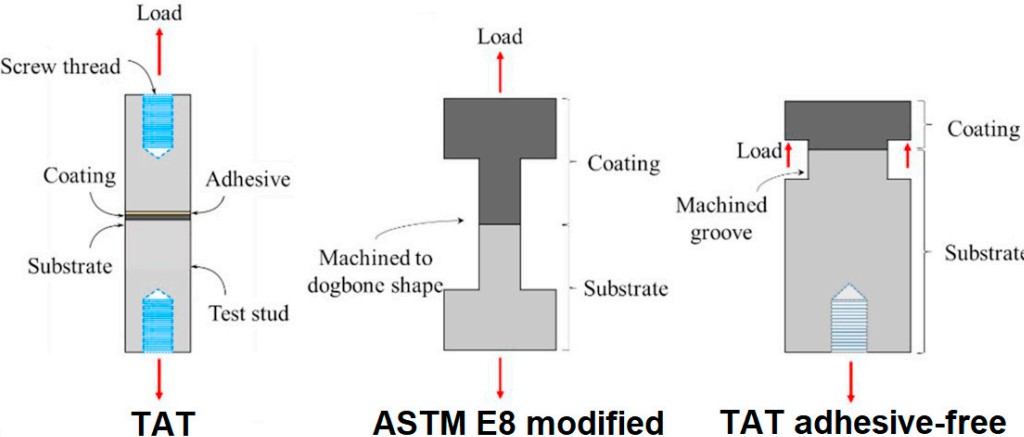

**Figure 12.** Schemes for CSAM adhesive testing. Reprinted with permission from Ref. [327], Elsivier, 2020.

The surface properties and the whole component quality are important for AM parts. Tensile testing has been done for different materials and fabrication strategies to measure the materials' strength and ductility [186,211,282]. Machining samples in different CSAM-ed part directions make the interpretation of mechanical isotropy possible, as performed by Yang et al. [200], Ren et al. [222], and Wu et al. [328]. The literature has presented the CSAM-ed mechanical resistance and ductility as a consequence of a good cohesion of particles, which TCT can easily measure. The TCT principle coats the circumference of two cylinders together head-to-head that are pulled in a universal testing machine. The stress or cohesion of particles is calculated as a relation between the loading collected and the coating thickness value, following the instructions of the EN 17,393:2020 standard [135].

Residual stress is crucial information for developing the CSAM as an industrial process, and the realization of the CSAM limitations is perhaps the main motivation behind the scholars' efforts to provide a reliable framework to study residual stresses in CS-ed deposits, initially by means of experimental and theoretical analyses, and later by finite element modeling. The residual stresses are divided into three types: the first order is macro-stresses homogeneous over multiple grains; the second order is micro-stresses over single grains; and the third order is micro-stresses in single grains, but with being inhomogeneous over the smallest areas such as unit cells [329]. Non-destructive diffraction measurement techniques for micro-stress have been used for CS-ed material, X-ray, and neutron diffraction. The first has a shallow penetration in the material, in order of micrometers, accrediting it just for superficial evaluation [258]. However, FEA was applied by Wang et al. [330] to simulate the residual stress along the CSAM-ed Cu part from X-ray diffractometry superficial residual stress results.

On the other hand, neutron diffraction penetrates the material in order of centimeters, but needs a long time exposition to achieve good results, in order of tens of minutes per measurement point [309]. Both methods are restricted to crystalline materials, and neutron diffraction has been studied for CSAM, as presented by Luzin et al. [331] and Vargas-Uscategui et al. [332] for Ti, Sinclair-Adamson et al. [333] for Cu, Loke et al. [334] for Al6061, and Boruah et al. [335] for Ti6Al4V. Despite being restricted to a few facilities worldwide and being an expensive technique, neutron diffraction has presented valid results in understanding the evolution of residual stress in CSAM deposition. In addition, it helps researchers to find new deposition strategies to reduce the regions with deleterious tensile fields.

Incremental Hole Drilling (IHD) is semi-destructive testing presented in the literature for the first-order residual stress measurement, which has been used for different materials and processes of fabrication for decades, including thermally sprayed coatings [336–338]. The IHD principle is based on drilling a small hole, <1 mm, into the material and collecting data about the deformations around the drilled hole using optical instruments or strain gauges. The material deformation or relaxation is related to the residual stress in the volume of the removed material through drilling [339], and the testing procedure is ruled by the ASTM E837-20 standard [340]. IHD is a technique routinely used for cast or rolled materials, and its use for CSAM promises high accuracy, easy sample preparation, and fast results. However, the literature still needs documents discussing the results and limitations and comparing IHD to other residual stress techniques, focusing on CSAM-ed bulks, which is a need to be filled by scholars.

In situ Coating Properties (ICP) measure the substrate curvature during and after deposition. The evolution of the sample curvature can be linked to the evolution of stresses in the thermally sprayed material using a variety of models [341]. Figure 13 shows an example of typical curves obtained by the ICP sensor, where there is evidence of the spraying time or deposition stress and the cooling time until room temperature, culminating in the residual stress. For HVOF sprayed coatings, normally, tensile residual stress is obtained, as indicated in Figure 13, with positive curvature values; however, for CS-ed coatings, the residual stress has negative curvature values, which is compressive. ICP has the advantages of being fast and not demanding the machining of samples, but it is limited to coatings, as shown by Sigh et al. [342], comparing ICP to X-ray for Inconel 718 coatings thinner than 1 mm, resulting in similar compressive residual stress values for both techniques. Furthermore, ICP does not apply to larger CSAM-ed parts, even though ICP results help the researchers optimize the CS parameters used for CSAM, mainly regarding improving adhesion.

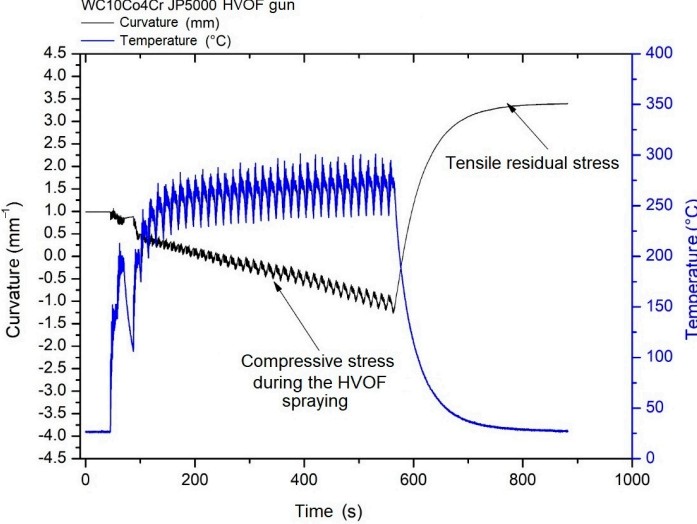

**Figure 13.** Typical curves obtained with ICP sensor.

Mechanical elements under cyclic loadings are subject to fatigue, reducing their life cycle. For CSAM-ed parts, Sample et al. [343] presented the influence of different CS properties, e.g., hardness, tensile properties, residual stress, etc., on the material fatigue performance. For CSAM-ed parts and CS-coated materials, different fatigue tests have been presented in the literature, which are designed and classified by the force or loading type: direct (axial) stress, plane bending, rotating beam, alternating torsion, or combined stress. Rotating beam and bending fatigue testing evaluate the parts exposed to revolutions under loading, such as axles, shafts, or wheels. Rotation bending exploits a rotating bending moment obtained through a rotating unbalanced mass, while the rotation beam places the load in the center of a supported sample at the ends. Applying CS as a coating improved the sample fatigue life by inserting compressive residual stress in the surface [258,344–348].

Using axial cyclic loading, three- and four-point bending fatigue testing have been presented in the literature. Xiong and Zhang [349] showed the improvement of mechanical resistance and fatigue life for an AZ91D Mg alloy after an LPCS-ed Al coating; Yamazaki, Fukuma, and Ohno [350] presented a low level of improvement in the fatigue life of CSAM repaired 316L samples, accrediting the repairing services for this material. Ševeček et al. [351] studied the benefits of CS-ed coatings on the Zircaloy-4 high-temperature fatigue life and displacement under cyclic loading. Considering the CSAM-ed bulk, Julien et al. [202] used compact tension specimens, following the ASTM E399-22 standard [352], to evaluate the fracture toughness ($K_{IC}$) of CSAM-ed Al6061. Wrought reference samples had much higher values than the CSAM-ed ones, 26.5 over 13.0 MPa·m$^{0.5}$, resulting from the CS-ed typical microstructure and the interparticular crack growth. A higher $K_{IC}$ reduction was presented by Kovarik et al. [221] for CSAM repairing Al, Ti, Ni, and Cu compared to rolled materials. Making the CSAM-ed parts have similar properties to bulks produced by traditional processes represents a challenge for CSAM's industrial application. Scholars have employed efforts to find solutions and possibilities to achieve solutions for this, such as post-treatments. Regarding the material fatigue life, Li et al. [353] proposed a probabilistic fatigue modeling for a GH4169 Ni alloy, using the weakest link theory applied to calculate the number of cycles to crack initiation. Similar modeling should be performed for CSAM-ed materials to compare how their microstructure defects and characteristics influence the material performance, deviating the experimental results from the mathematical and statistical model formulated.

### 3. Bibliometric Analysis

This section presents CSAM from an academic viewpoint, considering how the literature, scholars, and institutions cover the theme of CSAM. Bibliometric analysis has gained immense popularity in many research areas in the last decade due to being a powerful tool for interpreting the massive amount of data available nowadays, which, depending on the theme studied, may reach hundreds or even thousands of relevant documents [354]. Scholars use bibliometric analysis for different reasons, such as uncovering emerging publishing and journal performance trends, looking for investigation collaborators, or exploring the intellectual structure of a specific domain in the study [355]. The exciting use of bibliometric research is to identify knowledge gaps in the literature, helping the researchers to generate a novelty character in their future works filling these gaps.

It is not a new technique, the term bibliometrics was presented in the 1960s [356], and the evaluation of metrics regarding an area of interest in scientific publishing has been developed for more than a century [357]. Nowadays, in the big data era, this tool has been even more helpful in filtering and interpreting a large amount of information and data available for scholars. For AM, it is not different, and the bibliometric analysis has been related to the AM impact on business [358], on the supply chain [26], on industry 4.0 [359], AM-specific applications in orthopedics [360], or the general AM overview and trends [361], among others. Regarding CS, the literature presents the use of bibliometric analysis for a general overview comparing CS to other thermal spray processes [362–364];

however, there is a gap in the literature presenting the evolution of publishing focused on CSAM, or who the researchers and the institutions involved in this important theme are.

This work aims to understand the research status and development trends in the CSAM field, and identify the most relevant themes of study in the CSAM field, as there are some remarkable challenges. Therefore, it is important to carry out a bibliometric analysis that maps the current guidelines in this domain, which can inspire scholars in their future research lines and works. Furthermore, it gives them insights into the most active authors and journals that publish this theme and the countries that invest more in AM-related research. It provides a scientific cartography that reveals the dynamics and structure of scientific fields. For this purpose, a bibliometric analysis is conducted to map CSAM R&D trends.

### 3.1. Data Mining Strategy

The bibliometric data was extracted from the Scopus database using a query string containing keywords to search in the title, abstract, and keyword fields. Since this work has aimed to see the trend in publishing on CSAM over the last decade, the query string was refined to exclude articles published before 2012 and those in other languages. The following string retrieved more than 450 items as of 27 December 2022: TITLE-ABS-KEY (cold AND spray* AND additive AND manufactur*) OR TITLE-ABS-KEY (cold AND spray* AND 3d AND print*) AND PUBYEAR > 2011. These documents were subjected to further text cleaning and bibliometric analyses.

Due to their irrelevance to the studied theme, some articles were eliminated after a manual screen or database cleaning. The articles were limited to the subject area "materials science" OR "engineering" OR "physics and astronomy" OR "chemistry" OR "chemical engineering" OR "energy" OR "mathematics". The articles listed were carefully reviewed by reading their abstracts or full paper. The documents with an unclear relationship with the theme studied were eliminated, refining the results, resulting in the number of works for the statistical analysis being 439. Finally, the bibliometric analysis software VOSviewer was used to analyze the publications. VOSviewer is a software that graphically presents the bibliometric network mapping, which facilitates the interpretation of maps and data. The main networks are co-citation, bibliographic coupling, co-author, and/or co-word analysis. The authors and index keywords were selected for the co-occurrence analysis. VOSviewer identified many similar keywords, and to make the data more coherent, they were classified manually, such as "cold spray", "cold spraying", "cold gas dynamic spray", and "cold gas spray", which were merged to "cold spray".

### 3.2. Results and Discussions

Figure 14 presents the scientific productivity regarding the CSAM theme, limited to the last decade (2012–2022). The number of published documents each year indicates this technology's academic impact or interest by researchers, funding institutions, and journals. The number of documents per year significantly rose from 4 documents in 2012 to a maximum of 84 papers in 2022. This trend remained steady from 2020 and 2021, keeping around 81 publications per year. It is reasonable because the number of research groups researching CSAM and their productivity has not maintained the growth rate, despite the increasing number of researchers, groups, and equipment observed in the last decade [68].

Furthermore, implementing the LPCS process demands less investment because the equipment is less expensive, and the noise level during the operation is low [2,99]. Additionally, an LPCS gun is light and typically uses compressed air as the working gas and can be manipulated manually or using a small robot. However, to operate with HPCS equipment, a reasonable noise-insulated booth is demanded, as a facility for dozens of $N_2$ or He bottles [10], as well as the fact that the equipment costs of hundreds of thousands of dollars and a large size robot to support the gun, following the robot classification proposed by Dobra [365].

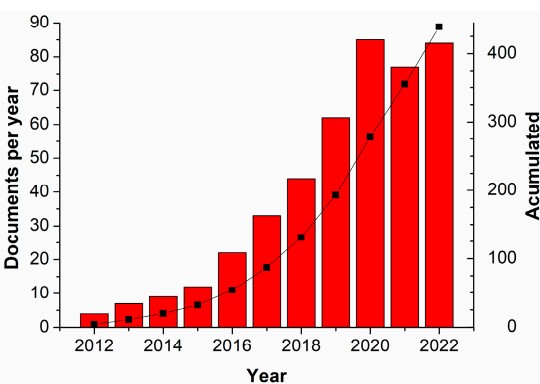

**Figure 14.** Year-wise publication of documents in CSAM field.

As seen in Figure 15, Halin Liao, who has an h-index of 61 in the Scopus database, is the researcher with more publications on the CSAM topic, with 35 documents. Liao has been a researcher at the Laboratoire Interdisciplinaire Carnot de Bourgogne/Université Bourgogne Franche-Comté (France) since 1994, and is co-author of 500 articles in diverse themes, such as materials characterization and performance, surface engineering, coatings, tribology, and corrosion, among others. Due to the relation of Liao with many other authors, his affiliation figures in the first position among the most important research groups, as seen in Figure 16, followed by Trinity College Dublin, due to the strong and numerous collaborations between Lupoi, Yin, and Chinese co-authors.

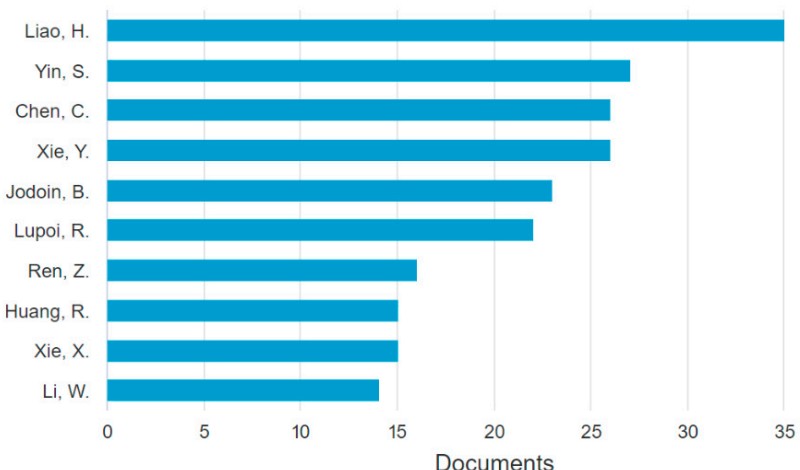

**Figure 15.** Number of documents per author.

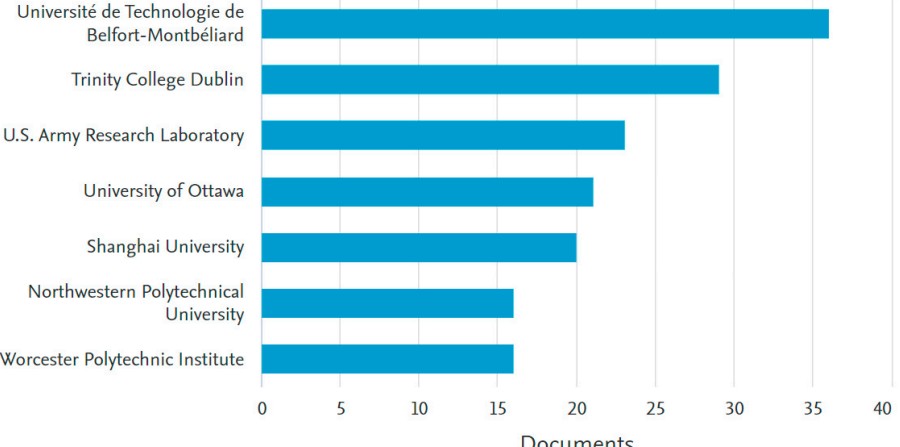

**Figure 16.** Number of documents by affiliation.

Most of the works published are in collaboration with researchers from Chinese institutions. For example, Shuo Yin, who has an h-index 36 in the Scopus database, is the second author in the number of published articles and worked with Halin Liao and Chaoyu Chen in France. Since 2015, he has been a researcher at Trinity College Dublin, where Rocco Lupoi, who has an h-index 31 in the Scopus database and is sixth on the publishers' list, develops his research too. Bertrand Jodoin, who has an h-index 33 in the Scopus database, is the fifth influential author in Figure 15 and works at the University of Ottawa (Canada).

China has the highest volume of documents published, mainly for collaborative works, as presented in Figure 17. China also leads this ranking, as it occurs in many other areas, due to the vast number of PhD students and active researchers at the Universities and research centers, as well as due to the massive amount of investments and R & D by governmental programs [366–369]. Even with an expressive high number of articles published by Chinese authors, the most cited articles in CSAM have only 6 Chinese figuring among the 25 co-authors enrolled in Table 4. However, the situation in the United States is worse because none of the important works presented in Table 4 has American co-authorship, for which the majority are Europeans. It reflects the importance given by the scientific community for the Chinese and American works, which could be by a lack of novelty seen in most of the hundreds of published works. Another consideration is that most works did not present new concepts but did an application and some important discussion on the concepts previously proposed by other original documents. Original documents or review articles have been cited more, as seen in Table 4. That article type is essential to consolidate the concepts but does not typically promote many citations, such as original or review articles [370,371]. It has caused a preoccupation by the Chinese institutions, which have looked at methodologies to make their work more recognized by the scientific community [372,373].

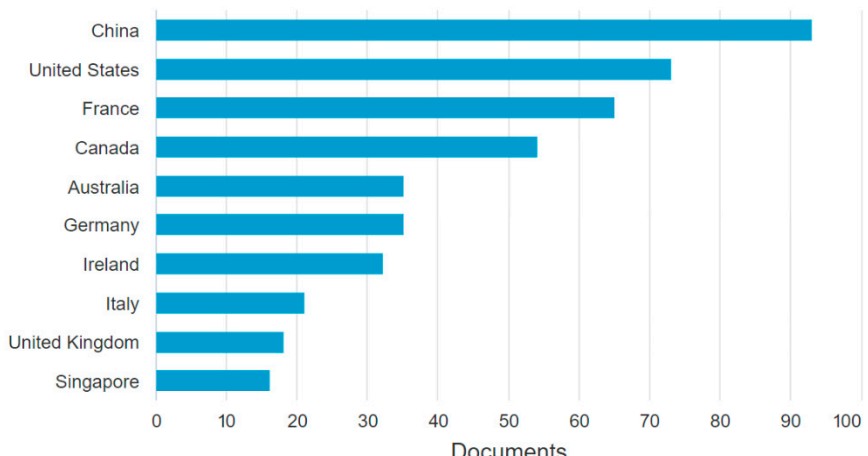

**Figure 17.** Number of documents by country.

**Table 4.** The articles most cited in CSAM theme.

| Title | Citations | Contributions and Goals | Ref. |
| --- | --- | --- | --- |
| Cold spraying—A material's perspective | 592 | An overview regarding the CS principles, ASI bonding mechanism, materials characteristics, and applications. | [77] |
| Cold spray additive manufacturing and repair: Fundamentals and applications | 372 | Summarizing and reviewing the CSAM-related work, comparing CSAM to fusion-based AM techniques, presenting the effects of HT on a CSAM-ed material's properties, and CSAM real applications. | [2] |

**Table 4.** *Cont.*

| Title | Citations | Contributions and Goals | Ref. |
|---|---|---|---|
| Solid-state additive manufacturing and repairing by cold spraying: A review | 235 | Summarizing and reviewing the CSAM-related work, different possibilities of CSAM application, alloys, process parameters, post-treatments, and their effects on CSAM-ed material mechanical properties. | [374] |
| Cold gas dynamic manufacturing: A non-thermal approach to freeform fabrication | 217 | Introducing the application of CS as an AM technique to produce freeform parts, comparing CSAM to other AM processes and CSAM strategies. | [169] |
| Cold gas dynamic spray additive manufacturing today: Deposit possibilities, technological solutions and viable applications | 212 | Presenting the evolution in investments in CSAM, adhesion and cohesion mechanisms for CSAM-ed material, listing materials and applications, characteristics, and industrial applications. | [68] |
| Potential of cold gas dynamic spray as additive manufacturing technology | 212 | Presenting the CSAM principles, geometric characteristics, and materials' properties, as well as the potential in using CSAM and its compatibility with other metal AM techniques. | [97] |

Collaborative works have characterized the articles and publishing in CSAM because of the mutual interests and the synergy in sharing equipment to develop the experiments and applying a kind of knowledge synergism to interpret the experimental results obtained. Regarding the authors' collaboration, the co-authorship relations were obtained by VOSviewer software, limiting the results to authors with more than ten articles published, reducing the total of 972 authors to the 16 presented in Figure 18. The circle size around the authors' names represents the number of articles in co-authorship, the color indicates a cluster of authors where the authors have more connections, and the line or link between the circles means the strength of their association; a thicker line means more collaborations. Chen and Xie are the leading authors in a cluster of Chinese cooperation, Yin and Lupoi are the most important authors in a cluster formed at Trinity College Dublin, and Liao is ahead of the French group. An interpretation of the map presented in Figure 18 is that its central persons are Liao, Xie, Chen, and Yin, indicating they act as bridges between the Chinese, Irish, and French institutions.

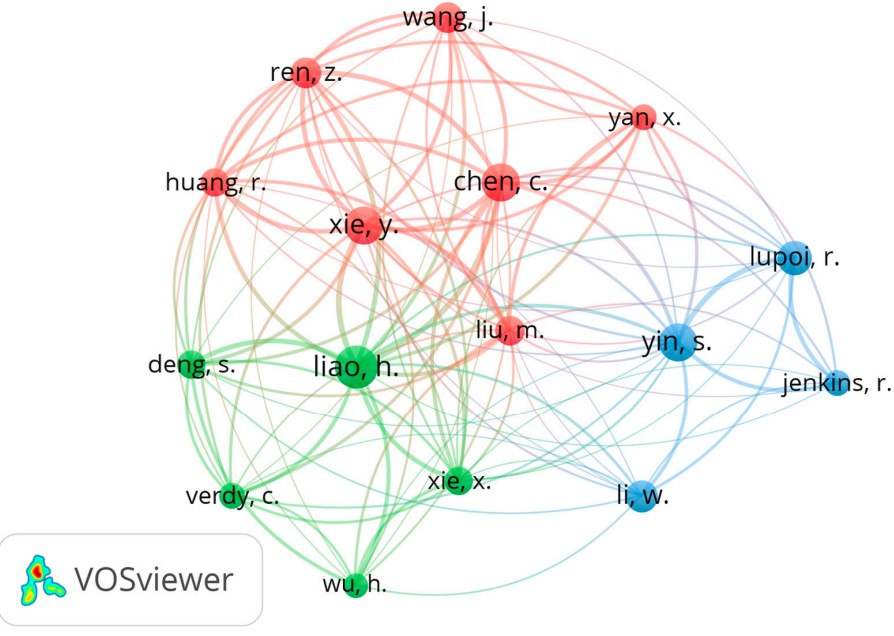

**Figure 18.** Authors' collaboration. Minimum of 10 articles per author.

The journal with more documents published, 48 articles, was the Journal of Thermal Spray Technology (JTST), a journal focused on surfaces, coatings, and films, justifying why the CS authors chose it to submit and publish their works. However, JTST places in the 70th percentile, or Q2, and has a cite score of 4.6 in the Scopus database. The second influential journal, with 29 papers, was Surface and Coatings Technology (SCT), older than JTST, in the 88th percentile, Q1, and with a cite score of 7.6 in the Scopus database. Open access journals have increased their contribution to CSAM publishing, attracting authors due to the faster publishing process and free access to the readers. Between the ten more relevant publishers, MDPI's journals Coatings, Materials, and Metals have 7, 10, and 10 documents published in the CSAM theme, respectively, from 2019 to 2022. MDPI's Metals has increased its relevance in the scientific community, publishing 6 documents only in 2021, reaching the 76th percentile, Q1, and 3.8 in the Scopus database.

Keywords represent the synthesis of the essential content of the documents, and their analysis aims to study the structure of the research related to the discipline. The analysis principle is based on the co-occurrence of keywords in the selected documents, revealing how closely they are connected in terms of the concepts they deal with, making it possible to understand the main themes of interest for the scholars. VOSviewer software identified more than 3000 keywords, and after a manual and critical evaluation, many of them were merged due to the similarity of their meaning. In addition, only keywords with at least 15 occurrences were considered, resulting in 70 keywords for the study, which are graphically presented in Figure 19b by their density, i.e., a darker and bigger circle represents more times the Keywords are listed: it results that "3d printing", "additive manufacturing", and "cold spraying" are the main terms, followed by "manufacturing processes", "additives", "coatings", and "microstructure" keywords.

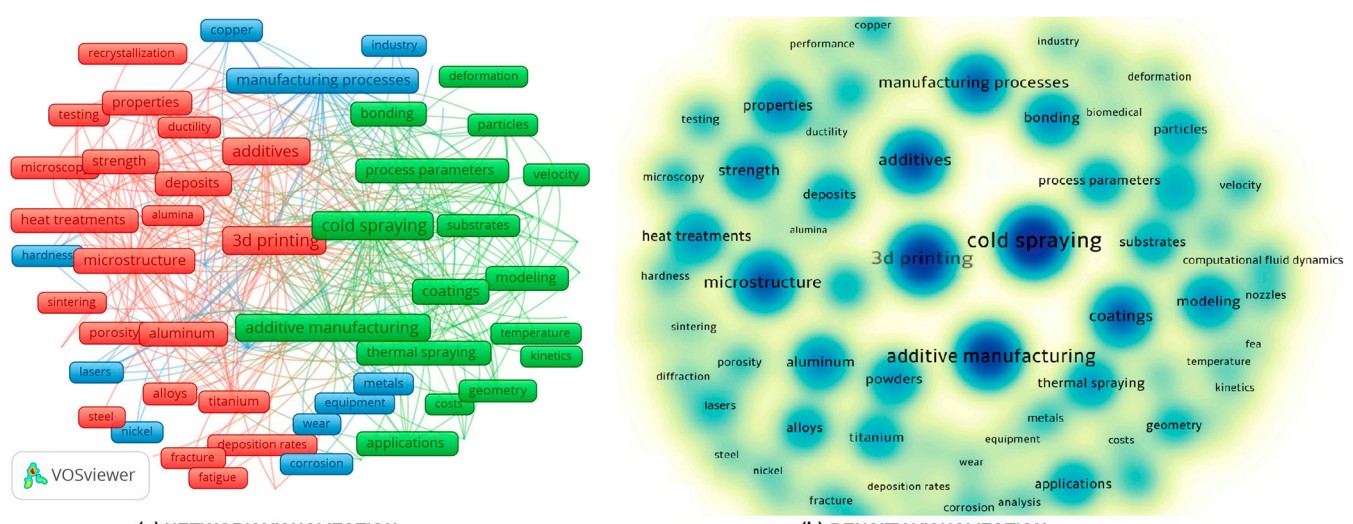

**Figure 19.** Co-occurrences keywords. Minimum 15 occurrences.

By analyzing the mapping network of co-occurrence, three clusters were formed, identified by the colors, Figure 19a. The clusters result from the link strength between the keywords, i.e., a stronger link means the same keywords group is used in more documents. VOSviewer resolution was set to 1.00 to avoid too small clusters. The smallest cluster, the blue one, has 18 items. The primary term is "manufacturing processes", which is a more generalist approach to making parts by CS, laser, and hybrid processes and is also linked to AM-processed materials, such as Cu, Ni, and composites. The other two clusters have the same number of items, 27; the green one has the main terms "cold spraying" and "additive manufacturing" that are strongly linked to "coatings", which makes perfect sense since CS has been developed prior for coating. The authors usually present these keywords together.

From the query string used for searching in the Scopus database, it was predictable that CS and AM would figure as the primary keywords for the papers.

A relation is understood from the materials linked to CSAM, which are Cu, Ti, and Al, as observed reading the articles, but these are presented in Figure 19a with stronger links in the blue cluster. Additionally, material properties and process parameters are highlighted because most works are experimental and present the materials' evaluation and testing. The red cluster presents the keyword "3d printing" linked to material properties, such as the microstructure, porosity, and strength. The keyword "3d printing" could be merged with "additive manufacturing", mixing the red and green clusters, following the AM nomenclatures in the literature. It makes perfect sense, considering the content of the papers that present these keywords. However, the "3d printing" term is a legacy from the AM of polymers and has been presented as a friendlier expression to AM non-experts. On the other hand, "rapid prototyping" had been a keyword widely used for AM [78,89], but in the last decade, it has been substituted for "3d printing" and "additive manufacturing", as indicated by Jemghili, Ait Taleb, and Khalifa [361]. It was confirmed by the VOSviewer keywords list andFigure 19, where "rapid prototyping" was not mentioned, indicating that this keyword has not been linked to CSAM.

## 4. Summary and CSAM Future Trends

This article briefly introduces CSAM, its characteristics, advantages over other AM processes, limitations, and some answers or alternatives to overtake them based on the literature. In addition, the paper presents challenges that still have to be overcome. Nevertheless, the innovation potential of this research field is outcoming, and new applications have emerged in different industrial fields, supporting the crescent number of publications dedicated to CSAM industrialization. Based on the state-of-art and interpretation of the most recent literature contents, some trends are listed:

- CSAM for repairing services, with its application on expensive components or damages that do not need extensive restoration [2]. Improving the CSAM-ed geometries control generates a hot topic for research, including geometry construction simulation, robot programming, and robot self-learning for an adaptive path, spraying angle, or gun displacement velocity. Research on this theme has been done by the Italian group of Politecnico di Milano [375], the Spanish group of Thermal Spray Center [144], and the Australian company Speed3D, among others;
- CSAM for hard materials, improving the CSAM-ed deposit adherence on materials such as Inconel, Ti6Al4V, HEA, or martensitic steels. For this, studies on the optimization of pre- and process-heating or CS parameters must be exploited. Some examples are using the expensive He as a working gas only for the first layers and $N_2$ for the others, the CS-SP process, or introducing HT between the layers to reduce the tensile residual stress on the CSAM/substrate interface and improve the adhesion and repairing quality;
- Improve CSAM-ed properties, reaching close or better than the wrought reference materials. As well as the well-established HT and HIP, new post-treatments have to be investigated in this theme. SPS presented good properties, but strict limitations in the geometries are feasible, requiring more flexibility for more complex geometries;
- CS hybrid systems consolidation, such as CS-SP or LACS, to avoid post-treatments and eliminate steps in the AM production chain [286]. Most studies are related to CS-ed coatings, promoting a better adhesion to the substrate and cohesion of particles, besides a low porosity. Therefore, CSAM hybrid systems' use is a hot topic to provide a good performance CSAM-ed parts;
- CSAM applied with other AM processes, optimizing the manufacturing chain to make the low complexity part areas by the fast CSAM process and dedicate the slower but more accurate laser process to the areas that demand more geometrical control. It is feasible because other AM techniques have increased their maturity as industrial

processes; however, this mixing of methods is a lack in R & D, which is a hot topic for scholars.

Regarding the bibliometric analysis, the literature characteristics and metrics were studied, collecting data from more than 420 documents published in the last decade for CSAM and related themes. The analysis covered several dimensions, including subject areas through keyword analysis, productive journals, the most influential authors, most cited documents, and referent affiliations and countries. The main results of the bibliometric analysis can be summarized as follows:

- A total of 56% of the total publications in the CSAM theme were registered during the last three years, indicating the increase of academic interest in this research field, considering that in 2010 the number of documents published was zero. The main topics actively explored in the papers were related to the processing parameters' optimization and other experiments focused on improving the CSAM-ed material's performance to make this process more industrially mature;
- China is the country with more documents published, followed by the United States and France, where the most relevant research group in CSAM is from, the Université de Technologie de Belfort-Montbéliard, which is the affiliation of Liao, the author with the most documents published. The publishing mapping presents a collaboration between Chinese and European institutions, signing for a fast CSAM industry maturity since the Chinese founding objectives are scientific development and even more advances in mass production;
- The current scenario of publication in CSAM points to a future consolidation of CSAM as an industrial technique, first for specific applications in high-cost components, such as multi-alloy nozzles for rockets in the aerospace industry or repairing expensive components, such as turbine blades or vanes. However, in the medium-term and long-term, CSAM applications tend to expand their use;
- "Costs" is the keyword that indicates a crucial point for CSAM advances. For the feedstocks, scholars have studied less expensive materials and improved DE, reaching more than 95% for some materials. A considerable challenge and trend for reducing processing costs and improving CSAM reliability is making the processing more independent of experts but easier for industrialization.

**Author Contributions:** Conceptualization, R.F.V.; funding acquisition, J.S. and I.G.C.; investigation, R.F.V. and A.G.; methodology, R.F.V. and A.G.; project administration, J.S. and I.G.C.; writing—original draft preparation, R.F.V., A.G., J.S. and V.A.; writing—review and editing, R.F.V., A.G., J.S. and V.A.; supervision, I.G.C. All authors have read and agreed to the published version of the manuscript.

**Funding:** The grant PID2020-115508RB-C21 and EIN2020-112379 was funded by MCIN/AEI/10.13039/501100011033 and, as appropriate, by "ERDF A way of making Europe", by "European Union NextGenerationEU/PRTR". A.G. and R.F.V. have AGAUR Ph.D. grants 2021 FISDU 00300, and 2020 FISDU 00305, respectively.

**Institutional Review Board Statement:** Not applicable.

**Informed Consent Statement:** Not applicable.

**Data Availability Statement:** The data presented in this study are available on request from the corresponding author.

**Conflicts of Interest:** The authors declare no conflict of interest. The funders had no role in the design of the study; in the collection, analyses, or interpretation of data; in the writing of the manuscript, or in the decision to publish the results.

## Abbreviations

The following abbreviations are used in this manuscript:

| | |
|---|---|
| AM | Additive Manufacturing |
| APS | Air Plasma Spray |
| ASI | Adiabatic Shear Instability |
| BB | Ball-Burnishing |
| BJ | Binder Jetting |
| CFD | Computational Fluid Dynamics |
| CR | Cold Rolling |
| CS | Cold Spray |
| CSAM | Cold Spray Additive Manufacturing |
| CS-SP | Cold Spray Shot Peening |
| DE | Deposition Efficiency |
| DMLM | Direct Metal Laser Melting |
| DMLS | Direct Metal Laser Sintering |
| EBM | Electron Beam Melting |
| EBSD | Electron Back Scattering Diffraction |
| EPP | Electric Pulsing Processing |
| FR | Flattening Ratio |
| FS | Flame Spraying |
| FSP | Friction Stir Processing |
| FSAM | Friction Stir Additive Manufacturing |
| FSW | Friction Stir Welding |
| GMAW | Gas Metal Arc Welding |
| GTAW | Gas Tungsten Arc Welding |
| HEA | High Entropy Alloy |
| HIP | Hot Isostatic Pressing |
| HPCS | High-Pressure Cold Spray |
| HR | Hot Rolling |
| HT | Heat Treatment |
| HVOF | High-Velocity Oxy-Fuel |
| ICP | In situ Coating Properties |
| IHD | Incremental Hole Drilling |
| JTST | Journal of Thermal Spray Technology |
| $K_{IC}$ | Fracture Toughness |
| LACS | Laser-Assisted Cold Spray |
| LMF | Laser Metal Fusion |
| LOM | Laminated Object Manufacturing |
| LPCS | Low-Pressure Cold Spray |
| MMC | Metal Matrix Composite |
| ME | Material Extrusion |
| MJ | Material Jetting |
| MMC | Metal Matrix Composite |
| MPCS | Medium-Pressure Cold Spray |
| PAW | Plasma Arc Welding |
| PEO | Plasma Electrolytic Oxidation |
| R&D | Research and Development |
| SCT | Surface and Coatings Technology |
| SD | Standoff Distance |
| SEM | Scanning Electron Microscopy |
| SL | Stereolithography |
| SLD | Supersonic Laser Deposition |
| SLM | Selective Laser Melting |
| SLS | Selective Laser Sintering |
| SP | Shot Peening |

| SPS | Spark Plasma Sintering |
|---|---|
| TAT | Tensile Adhesion Testing |
| TCT | Tubular Coating Tensile |
| TMT | Thermo-Mechanical Treatment |
| UAM | Ultrasonic Additive Manufacturing |
| UTS | Ultimate Tensile Strength |
| $V_{cr}$ | Critical Velocity |
| $V_{er}$ | Erosion Velocity |
| WAAM | Wire Arc Additive Manufacturing |
| WCI | Waterjet Cavitation Impact |

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
