# Peer review of "A Review of Advances in Cold Spray Additive Manufacturing"

_coatings, doi:10.3390/coatings13020267_

Round 1

Reviewer 1 Report

This paper reviews Cold Spray Additive Manufacturing (CSAM) literature, emphasizing the main concepts, advantages, and drawbacks of the method. The bibliographic analysis with the aid of VOSviewer software underlined the trend of the research in the field and the existing increasing interest in the world.

Covering 362 published papers in CSAM, the review makes an important contribution. 

My suggestion for the authors is to include a flowchart of the items discussed in this paper at the end of the introduction section. Also, two other flowcharts should underline the main advantages and drawbacks of the method, and the possibilities to overcome these inconveniences.

All over the paper are wrong hyperlinks to the cited references, and this issue should be solved. 

Author Response

The authors thank the reviewer's comments and suggestions. The authors contest the reviewer's requests:

1) My suggestion for the authors is to include a flowchart of the items discussed in this paper at the end of the introduction section. Also, two other flowcharts should underline the main advantages and drawbacks of the method, and the possibilities to overcome these inconveniences.

R: The authors agree that the flowcharts and schemes should help the reader better interpret the themes presented in the paper. The authors inserted two schemes in the manuscript.

2) All over the paper are wrong hyperlinks to the cited references, and this issue should be solved.

R: The authors reviewed the document and edit the hyperlinks.

Reviewer 2 Report

Dear Author,

The article presents an interesting point of view on the production of products using the cold spray method in the process of manufacturing models with selected 3D printing technologies. However, it requires a few changes, which are listed below.

1. Introduction should be extended with the progress that has been made in optimizing 3D printing, which also affects the mass of models, overall material consumption and energy consumption, which has a large impact on ecology. It proposes an analysis of the literature in this area, for example: The Influence of Selected Selective Laser Sintering Technology Process Parameters on Stress Relaxation, Mass of Models, and Their Surface Texture Quality, DOI10.1089/3dp.2019.0036 A Comparative Study of the Mechanical Properties of FDM 3D Prints Made of PLA and Carbon Fiber-Reinforced PLA for Thin-Walled Applications, DOI10.3390/ma14227062

2. There is no reference to literature on page 2 - errors, this problem occurs later in the text.

3. Does Figure 2 represent all the technologies in this area?

4. Do you mean DMLM or DMLS?

5. How was the data for table 1 selected in point 2.1?

6. Is figure 6 for self-made samples? there is no connection to the literature.

7. I propose to discuss Figure 9 in detail.

8. Was the literature analysis carried out only in the scopus database or in other theses, for example, web of science? Which base do the results presented in Figure 12 refer to?

9. In the last chapter, I recommend to discuss the topic of other 3D and CS printing technologies in this area.

Kind regards,

Reviewer

Author Response

The authors thank the reviewer for the comments and suggestions. The authors reply the items listed by the reviewer:

  1. Introduction should be extended with the progress that has been made in optimizing 3D printing, which also affects the mass of models, overall material consumption and energy consumption, which has a large impact on ecology. It proposes an analysis of the literature in this area, for example: The Influence of Selected Selective Laser Sintering Technology Process Parameters on Stress Relaxation, Mass of Models, and Their Surface Texture Quality, DOI10.1089/3dp.2019.0036 A Comparative Study of the Mechanical Properties of FDM 3D Prints Made of PLA and Carbon Fiber-Reinforced PLA for Thin-Walled Applications, DOI10.3390/ma14227062

R: The authors thank for the suggestion. The authors studied critically the articles suggested and inserted in them in the manuscript.

  1. There is no reference to literature on page 2 - errors, this problem occurs later in the text.

R: The manuscript was reviewed, and the hyperlinks were rectified.

  1. Does Figure 2 represent all the technologies in this area?

R: The authors edited Figure 2 to reduce the number of images, keeping only those that help the readers to understand what CSAM makes nowadays and what is made by other metal AM techniques. One of the objectives of presenting parts made by other methods is to show that CSAM is only a substitute in some AM applications. For example, there are parts for which welding or laser is the best solution, e.g., more complex geometries or very large parts. The Figure caption presents the process used for each part.

  1. Do you mean DMLM or DMLS?

R: The article presents they two, Direct Metal Laser Melting (DMLM) and (Direct Laser Sintering (DMLS).

  1. How was the data for table 1 selected in point 2.1?

R: Table 1 summarizes the critical velocity and erosion velocity for CS of the most used materials for CSAM. The section “2.1 Cold Spray Parameters” discusses the sprayed particle velocity and the effect of CS parameters on this velocity, which influences the material consolidation or bonding.

  1. Is figure 6 for self-made samples? there is no connection to the literature.

R: Figure 6 results from the authors' experiments at the CPT (University of Barcelona) laboratories. This Figure had yet to be published elsewhere.

  1. I propose to discuss Figure 9 in detail.

R: The authors inserted a discussion about the Figure 9:

“Figure 9 shows CSAM-ed Ni/FeSiAl in as-sprayed and HT-ed conditions, Figure 9 (a) and (b), respectively. HT promoted a recrystallization, grain coarsening, and phase transformation in the material, as interpreted from the size of each colored area, which is more significant in Figure 9 (b). This image has fewer areas without a defined atomic lattice plane of the crystalline structure, as seen in the as-sprayed condition as dark green. These dark green areas represent patterns uninterpreted by the detector, which is related to the severe particle deformation during the CS deposition. The improvement in this indexing rate from 78 to 90% represents the recrystallization phenomena from HT post-treatment [294].”

  1. Was the literature analysis carried out only in the scopus database or in other theses, for example, web of science? Which base do the results presented in Figure 12 refer to?

R: The bibliometric analysis was done using only the Scopus database. In the section “3.1. Data Mining Strategy” this is specified: “The bibliometric data was extracted from the Scopus database”.

  1. In the last chapter, I recommend to discuss the topic of other 3D and CS printing technologies in this area.

R: The authors reviewed the summary section and considered the evolution of other AM processes, as suggested by the reviewer.

Reviewer 3 Report

In this manuscript, the authors thoroughly review the advances in cold spray additive manufacturing. The review is comprehensive, with 362 references, well written, and covers the description of cold spray processes, process parameters, process challenges, and characterization techniques of manufacturing parts. Moreover, the results of the bibliometric analysis are discussed, and the contributions of research groups and scientists are identified.

The comments related to the present manuscript are summarized below.

 1.      Consider adding graphical illustrations of the relationships between the powder properties, process parameters, and manufactured part properties.

2.      Consider adding the review of papers studying the effects of powder properties on powder flowability to ensure a reliable powder supply.

3.      Correct the errors with figure references in the text.

Author Response

The authors thank thr reviewer comments and suggestions. The authors replay the reviewer listed items:

  1. Consider adding graphical illustrations of the relationships between the powder properties, process parameters, and manufactured part properties.

R: The authors agree that powders are extremally important for CS deposition and inserted a text summarizing the powder/deposition relationship.

  1. Consider adding the review of papers studying the effects of powder properties on powder flowability to ensure a reliable powder supply.

R: The authors added more discussion to the powders review previously presented.

“CS almost always uses conventional powders as feedstock materials developed for Air Plasma Spray (APS), High-Velocity Oxy-Fuel (HVOF), or laser processes in a spherical and finer particle size range at best. Various techniques are available to pro-duce metallic powders, which are chosen by the chemical composition, characteristics, and/or properties required for the powder [147]. For the CS, the particles' metallurgical, morphological, and physico–chemical characteristics influence the spraying success and material performance [85]. Since CS does not promote recrystallization during the deposition, a deposit with refined microstructure is obtained by selecting a small grain-size feedstock powder. It improves the mechanical properties; however, a larger gain size promotes more ductility to the particle. Using HT to reach the ideal powder microstructure was an alternative presented by Poirier et al. [148] for CS H13 tool steel, and by Story and Brewer [149] for aluminum alloys, resulting in a DE increasing from 35 to 60% and from 70 to 90% to Al7075 and Al6061, respectively. By modeling and experimental results, Silvello et al. [86] summarized the relationship between powder characteristics, CS process parameters, and the CS sprayed material properties. Table 2 presents coefficients for the model proposed using modeFRONTIER software, in which negative values represent inverse input/output relationships. It is noticed that the particle diameter and hardness influence the CS sprayed material characteristics, highlighting the porosity, to which is attributed some CS drawbacks, like short fatigue life.

Table 2. Correlation behavior among the different input/output for CS [86].

Input /
Output

Particle
diameter

Particle
hardness

Gas
Pressure

Gas
temperature

Particle
velocity

Deposit
hardness

Porosity

DE

FR

Particle
diameter

1

0

0

0

−0.431

−0.187

−0.213

0.104

0.097

Particle
hardness

0

1

0

0

0

0.935

0.109

0

−0.324

Gas
pressure

0

0

1

0

0.594

0.417

−0.682

0.768

0.804

Gas
temperature

0

0

0

1

0.498

0.297

−0.471

0.592

0.897

Particle
velocity

−0.431

0

0.594

0.498

1

0.682

−0.734

0.803

0.817

Deposit
hardness

−0.187

0.935

0.417

0.297

0.682

1

0

0

−0.352

Porosity

−0.213

0.109

−0.682

−0.471

−0.734

0

1

0

−0.819

DE

0.104

0

0.768

0.592

0.803

0

0

1

0

FR

0.097

−0.324

0.804

0.897

0.817

−0.352

−0.819

0

1

CS powders must be characterized before spraying, measuring their particle size distribution by ASTM B214 standard [150], a sieving separation of the larger and smaller particles, or the laser scattering, classifying the particle size distribution by measuring the laser illuminated flowing particles. The powder flowability has been measured by the time elapsed to flow a certain powder mass through a certified Hall flowmeter, following the ASTM B213-20 standard [151], which is used to measure the powder apparent density, as indicated by the ASTM B212-21 standard [152]. A previous characterization of the powder is imperative since powders’ flowrate higher than 1 g·s–1 tend to build up and block the gas flow in the nozzles for LPCS [147]. For HPCS, Vaz et al. [133] presented the flowability for different 316L, resulting in 9 and 17 g·s–1, for the irregular and spherical shapes, respectively. This powder characteristic impacted the CS powder feeding, which were 0.43 and 0.55 g·s–1 for the irregular and spherical shapes, respectively. By machining learning, Valente et al. [153] show how to predict a novel powder flowability on a per-particle basis, which can help scholars develop their alloys and powders for CSAM.

An irregular shape of particles does not necessarily result in a coating or CSAM part with worse properties [154–156]. The high deformation of the CS sprayed particles at the impact can act as compensation for their shape irregularity and even for the particle size distribution, which enables using coarse particles, as presented by Singh et al. [155], who obtained similar material’s strength by coarse and fine Cu particles. CS 316L coatings using water-atomized powders, which had an irregular shape, presented corrosion behavior and wear-resistance performance very similar to, or even better than, the coatings obtained with spherical gas-atomized powders [133], indicating the viability of using a lower-cost raw material for CS, since the 316L gas-atomized powders are more expensive than the water-atomized ones. Wong et al. [154] obtained very similar porosity values (3.0±0.5%), DE (100%), and hardness (200±10 HV) for CS Ti coatings employing irregular and spherical shape powders, but considering coating quality, the authors suggested the spherical medium-sized powder the best CS option. For Ti6Al4V, spherical particles presented higher hardness and cohesive strength than a very irregular powder obtained by Armstrong process, as shown by Munagala et al. [157]. The powder size distribution influences the CS sprayed particles’ velocity; smaller particles reach higher velocities than bigger and weightier ones, as presented in a simulation performed for 5, 25, and 50 μm Al particles. The first one resulted in a velocity higher than 600 m·s–1, but the last one was lower than 500 m·s–1 [158]. For CS sprayed Cu particles, small particles, 5 µm, reached a velocity of 700 m·s–1, while big particles, 90 µm, accelerated up to 300 m·s–1. Bagherifard et al. [120] presented 316L fine powder, –29+12 μm, with higher spraying velocity than coarse particles, –45+19 μm, which resulted in a material with higher particle deformation, mechanical properties, and electrical conductivity. Meantime, Vcr is dependent on the particle size, and smaller particles have much higher Vcr than the bigger ones, resulting in even under high velocity, small particles may not bond, and an optimum size range is achieved for each material, which is generally between 10 and 60 μm. Improving the temperature of particles, Vcr, is reduced, revealing the need to improve the CS working gas temperature to increase the temperature of smaller particles and the velocity of bigger particles [134,159,160]; however, higher gas temperatures put the equipment in an undesired condition, overloading it and promoting nozzle clogging.”

  1. Correct the errors with figure references in the text.

R: The authors reviewed the document and edit the hyperlinks.

Reviewer 4 Report

In this paper, a literature is presented to improve CSAM materials’ quality, properties, and possibilities of use. It reviews these advances in the last decade, considering new materials, process parameters optimization, post-treatments, and hybrid processing. The paper has some interesting results that could make it publishable in the journal of Coatings after the following major revisions:

1- Please check the English language of the paper thoroughly.

2-Move these two sentences of the abstract to the introduction:

“Additive Manufacturing (AM) has been an industrial revolution in the last decades, starting with producing polymeric parts and advancing to metallic components. Many alloys and methods have been studied, some more industrially mature and others in a developing stage. “  

3-Define in the abstract what parameters were assessed.

4-Introduciton should be strengthened. Some old references are used. To modify this section the following documents can be consulted:

-(2021). Microstructure and mechanical properties of ultrasonic spot welding TiNi/Ti6Al4V dissimilar materials using pure Al coating. Journal of Manufacturing Processes, 64, 473-480. doi: https://doi.org/10.1016/j.jmapro.2021.02.009

-(2022). Water jet impact damage mechanism and dynamic penetration energy absorption of 2A12 aluminum alloy. Vacuum, 206, 111532. doi: https://doi.org/10.1016/j.vacuum.2022.111532

-(2023). Microstructural understanding of the oxidation and inter-diffusion behavior of Cr-coated Alloy 800H in supercritical water. Corrosion Science, 211, 110910. doi: https://doi.org/10.1016/j.corsci.2022.110910

5-Figure 2 is a bit busy. If possible, reduce the number of photos in this figure.

7-In figure 9, use a and b in the caption.

8-Correct the caption in figure 12.

9-Consult the following references in the discussion section:

-(2023). Effect of heat treatment process on the micro machinability of 7075 aluminum alloy. Vacuum, 207, 111574. doi: https://doi.org/10.1016/j.vacuum.2022.111574

-(2022). Probabilistic fatigue modelling of metallic materials under notch and size effect using the weakest link theory. International Journal of Fatigue, 159, 106788. doi: https://doi.org/10.1016/j.ijfatigue.2022.106788

(2022). Transient thermomechanical analysis of micro cylindrical asperity sliding contact of SnSbCu alloy. Tribology international, 167, 107362. doi: 10.1016/j.triboint.2021.107362

10-Summary is too long and tedious. If possible, make it as bullet points. 

Author Response

The authors thank the reviewer's comments and suggestions. The authors reply the reviewer listed items:

1- Please check the English language of the paper thoroughly.

R: The text was reviewed.

2-Move these two sentences of the abstract to the introduction:

“Additive Manufacturing (AM) has been an industrial revolution in the last decades, starting with producing polymeric parts and advancing to metallic components. Many alloys and methods have been studied, some more industrially mature and others in a developing stage”.

R: OK, text moved.

3-Define in the abstract what parameters were assessed.

R: The authors reviewed the Abstract section and inserted some information regarding the methodology to make the literature review.

“Additive Manufacturing (CSAM) produces freeform parts by accelerating powder particles at supersonic speed, which impacting against a substrate material trigger a process to consolidate the CSAM part by bonding mechanisms. The literature has presented scholars’ efforts to improve CSAM materials’ quality, properties, and possibilities of use. This work is a review of the CSAM advances in the last decade, considering new materials, process parameters optimization, post-treatments, and hybrid processing. The literature considered includes articles, books, standards, and patents, which were selected by the relevance with the CSAM theme. In addition, this work contributes to compiling important information from the literature that and presents how CSAM has advanced fast in diverse sectors and applications. Another approach presented is the academic contributions by a bibliometric review, considering showing the most relevant contributors, authors, institutions, and countries during the last decade for the CSAM development research. Finally, it is presented a trend for the future of CSAM, its challenges, and barriers to be overcome.”

4-Introduciton should be strengthened. Some old references are used. To modify this section the following documents can be consulted:

-(2021). Microstructure and mechanical properties of ultrasonic spot welding TiNi/Ti6Al4V dissimilar materials using pure Al coating. Journal of Manufacturing Processes, 64, 473-480. doi: https://doi.org/10.1016/j.jmapro.2021.02.009

-(2022). Water jet impact damage mechanism and dynamic penetration energy absorption of 2A12 aluminum alloy. Vacuum, 206, 111532. doi: https://doi.org/10.1016/j.vacuum.2022.111532

-(2023). Microstructural understanding of the oxidation and inter-diffusion behavior of Cr-coated Alloy 800H in supercritical water. Corrosion Science, 211, 110910. doi: https://doi.org/10.1016/j.corsci.2022.110910

R: The authors thank the reviewer for the suggested references. They were read critically and added to the manuscript. The older references used in the manuscript are important to interpret the evolution of the techniques and how the materials and problems considered for the CSAM process have been studied previously for other methods, which indicates the insertion of CSAM as an alternative for these other techniques, such laser, welding, and others.

Few more recent references were added to the manuscript:

Chen, C.; Su, H.; Wang, X.; Liu, Y.; Zhao, L.; Wei, X.; Zhao, Y.; Pan, J.; Qiu, X. Impact-Induced Bonding Process of Copper at Low Velocity and Room Temperature. Mater. Des. 2023, 111603, doi:10.1016/j.matdes.2023.111603.

Zhang, P.; Liu, J.; Gao, Y.; Liu, Z.; Mai, Q. Effect of Heat Treatment Process on the Micro Machinability of 7075 Aluminum Alloy. Vacuum 2023, 207, 111574, doi:10.1016/j.vacuum.2022.111574.

Mayer, A.R.; Bertuol, K.; Siqueira, I.B.A.F.A.F.; Chicoski, A.; Váz, R.F.; de Sousa, M.J.; Pukasiewicz, A.G.M.M. Evaluation of Cavitation/Corrosion Synergy of the Cr3C2-25NiCr Coating Deposited by HVOF Process. Ultrason. Sonochem. 2020, 69, 1–9, doi:10.1016/j.ultsonch.2020.105271.

Soyama, H. Cavitation Peening: A Review. Metals (Basel). 2020, 10, 270, doi:10.3390/met10020270.

Zhang, P.; Liu, Z.; Yue, X.; Wang, P.; Zhai, Y. Water Jet Impact Damage Mechanism and Dynamic Penetration Energy Absorption of 2A12 Aluminum Alloy. Vacuum 2022, 206, 111532, doi:10.1016/j.vacuum.2022.111532.

Soyama, H.; Okura, Y. The Use of Various Peening Methods to Improve the Fatigue Strength of Titanium Alloy Ti6Al4V Manufactured by Electron Beam Melting. AIMS Mater. Sci. 2018, 5, 1000–1015, doi:10.3934/matersci.2018.5.1000.

5-Figure 2 is a bit busy. If possible, reduce the number of photos in this figure.

R: The authors eliminated the less relevant images, keeping only the images that helps the readers to understand what AM is capable to produce and how complex or simple the parts can be.

7-In figure 9, use a and b in the caption.

R: Figure and caption edited.

8-Correct the caption in figure 12.

R: Figure caption edited.

9-Consult the following references in the discussion section:

-(2023). Effect of heat treatment process on the micro machinability of 7075 aluminum alloy. Vacuum, 207, 111574. doi: https://doi.org/10.1016/j.vacuum.2022.111574

-(2022). Probabilistic fatigue modelling of metallic materials under notch and size effect using the weakest link theory. International Journal of Fatigue, 159, 106788. doi: https://doi.org/10.1016/j.ijfatigue.2022.106788

-(2022). Transient thermomechanical analysis of micro cylindrical asperity sliding contact of SnSbCu alloy. Tribology international, 167, 107362. doi: 10.1016/j.triboint.2021.107362

R: The authors thank for the recommendations. After reading critically the papers, the pertinent references were inserted in the review work.

10-Summary is too long and tedious. If possible, make it as bullet points.

R: The authors agree with the Reviewer that some information were redundant in the section and the Summary and Future Trends section was edited and rewrote:

“This article briefly introduced the CSAM, its characteristics, advantages over other AM processes, limitations, and some answers or alternatives to overtake them based on the literature. Besides that, the paper presented challenges that still have to be overcome. Nevertheless, the innovation potential of this research field is outcoming, and new applications have emerged in different industrial fields, supporting the crescent number of publications dedicated to CSAM industrialization. Based on the state-of-art and interpretation of the most recent literature contents, some trends are listed:

  • CSAM for repairing services, with application on expensive components or dam-ages that do not need extensive restoration [2]. Improving the CSAM made geometries control generates a hot topic for research, including geometry construction simulation, robot programming, and robot self-learning for an adaptative path, spraying angle, or gun displacement velocity. Research on this theme has been done by the Italian group of Politecnico di Milano [353], the Spanish group of Thermal Spray Centre [145], and the Australian company Speed3D, among others;
  • CSAM for hard materials, improving the CSAM deposit adherence on materials like Inconel, Ti6Al4V, HEA, or martensitic steels. For this, studies on the pre- and process-heating or CS parameters optimization must be exploited. Some examples are using the expensive He as working gas only for the first layers and N2 for the others, CS-SP process, or introducing HT between the layers to reduce the tensile residual stress on the CSAM/substrate interface, improve the adhesion and repairing quality;
  • Improve CSAM properties, reaching close or better ones than the wrought reference materials. Besides the well-established HT and HIP, new post-treatments have to be investigated in this theme. SPS presented good properties, but strict limitations in the geometries are feasible, requiring more flexibility for more com-plex geometries;
  • CS hybrid systems consolidation, like CS-SP or LACS, to avoid post-treatments and eliminate steps in the AM production chain [267]. Most studies are related to CS coatings, promoting a better adhesion to the substrate and cohesion of particles, besides a low porosity. Therefore, CSAM hybrid systems use is a hot topic to pro-vide good performance CSAM parts.

Regarding the bibliometric analysis, the literature characteristics and metrics were studied, collecting data from more than 420 documents published in the last decade for the CSAM and related themes. The analysis covered several dimensions, including subject areas through keyword analysis, productive journals, most influential authors, most cited documents, and referent affiliations and countries. The main results of the bibliometric analysis can be summarized as follows:

  • 56% of the total publications in the CSAM theme were registered during the last three years, indicating the increase of academic interest in this research field, considering that in 2010 the number of documents published was zero. The main topics actively explored in the papers were related to processing parameters optimization and other experiments focused on improving the CSAM material performance to make this process more industrially mature.
  • China is the country with more documents published, followed by the United States and France, from where is the most relevant research group in CSAM, the Université de Technologie de Belfort-Montbéliard, which is the affiliation of Liao, the author with more documents published. The publishing mapping presents a collaboration between Chinese and European institutions, signing for a fast CSAM industry maturity since the Chinese founding objectives are scientific development and even more advances in mass production;
  • The current scenario of publication in CSAM points to a future consolidation of CSAM as an industrial technique, first for specific applications in high-cost components, like multi-alloy nozzle for rockets in the aerospace industry or repairing expensive components, such as turbine blades or vanes. But in the medium term and long term, CSAM applications tend to expand their use;
  • “Costs” is the keyword that indicates a crucial point for CSAM advances. For the feedstocks, scholars have studied less expensive materials and improved DE, reaching more than 95% for some materials. A considerable challenge and trend for reducing processing costs and improving CSAM reliability are making the processing more independent of experts but easier for industrialization.”

Round 2

Reviewer 4 Report

The paper can now be accepted in its revised format. 

Author Response

The authors are glad to achieve the Reviewer's expectations.